# CRISPR-based targeted haplotype-resolved assembly of a megabase region

Taotao Li[1,2,9], Duo Du[1,2,9], Dandan Zhang[1,2,9], Yicheng Lin[1,2,9], Jiakang Ma[1,2], Mengyu Zhou[1,2], Weida Meng[1,2], Zelin Jin[1,2], Ziqiang Chen[1,2], Haozhe Yuan[1,2], Jue Wang[1,2], Shulong Dong[1,2], Shaoyang Sun[3], Wenjing Ye[4], Bosen Li[3], Houbao Liu[5], Zhao Zhang[3], Yuchen Jiao[6], Zhi Xie[7], Wenqing Qiu ®[1,8] ✉ & Yun Liu ®[1,2] ✉

Constructing high-quality haplotype-resolved genome assemblies has substantially improved the ability to detect and characterize genetic variants. A targeted approach providing readily access to the rich information from haplotype-resolved genome assemblies will be appealing to groups of basic researchers and medical scientists focused on specific genomic regions. Here, using the 4.5 megabase, notoriously difficult-to-assemble major histocompatibility complex (MHC) region as an example, we demonstrated an approach to construct haplotype-resolved assembly of the targeted genomic region with the CRISPR-based enrichment. Compared to the results from haplotype-resolved genome assembly, our targeted approach achieved comparable completeness and accuracy with reduced computing complexity, sequencing cost, as well as the amount of starting materials. Moreover, using the targeted assembled personal MHC haplotypes as the reference both improves the quantification accuracy for sequencing data and enables allele-specific functional genomics analyses of the MHC region. Given its highly efficient use of resources, our approach can greatly facilitate population genetic studies of targeted regions, and may pave a new way to elucidate the molecular mechanisms in disease etiology.

The recent advances in constructing high-quality haplotype-resolved genome assemblies have substantially improved sensitivity in detecting and characterizing genetic variants[1,2], and will greatly advance our understanding of the human genome. When parent–child trio information is available, the trio-binning approach permits the construction of haplotype-resolved genome assemblies for offspring individuals[3–5]. Alternatively, the PacBio high-fidelity (HiFi) long-read sequencing, coupled with strand-specific sequencing technologies, provides another option to assemble the diploid genome without the need for pedigree information[1,6–9]. However, both approaches to achieving

[1]MOE Key Laboratory of Metabolism and Molecular Medicine, Department of Biochemistry and Molecular Biology, School of Basic Medical Sciences and Shanghai Xuhui Central Hospital, Fudan University, Shanghai, China. [2]State Key Laboratory of Medical Neurobiology and MOE Frontiers Center for Brain Science, Institutes of Brain Science, Fudan University, Shanghai, China. [3]MOE Key Laboratory of Metabolism and Molecular Medicine, Department of Biochemistry and Molecular Biology, School of Basic Medical Sciences, Fudan University, Shanghai, China. [4]Division of Rheumatology and Immunology, Huashan Hospital, Fudan University, Shanghai, China. [5]Department of General Surgery, Zhongshan Hospital, Fudan University, Shanghai, China. [6]State Key Laboratory of Molecular Oncology, National Cancer Center/National Clinical Research Center for Cancer/Cancer Hospital, Chinese Academy of Medical Sciences and Peking Union Medical College, Beijing, China. [7]State Key Laboratory of Ophthalmology, Zhongshan Ophthalmic Center, Sun Yat-sen University, Guangzhou, China. [8]Human Phenome Institute, Zhangjiang Fudan International Innovation Center, Fudan University, Shanghai, China. [9]These authors contributed equally: Taotao Li, Duo Du, Dandan Zhang, Yicheng Lin. ✉e-mail: qiuwq@fudan.edu.cn; yliu39@fudan.edu.cn

high-quality haplotype-resolved de novo genome assemblies require substantial large amounts of starting materials, extensive computing resources, and high levels of sequencing costs. A targeted approach providing ready access to the rich information from haplotype-resolved genome assemblies will be appealing to groups of basic researchers and medical scientists, who are focused on specific genomic regions for their functional relevance.

Recently, CRISPR/Cas9 starts to be adopted as a tool for the enrichment of targeted genomic regions[10–14]. Compared with probe-based DNA-enrichment strategies, this approach only needs to know the DNA sequences flanking the targeted genomic region, making it an ideal tool for the unbiased enrichment of targeted regions, even with high polymorphisms. Moreover, it is compatible with long-read sequencing technologies[10–14] required for haplotype-resolved assemblies. We envisioned that adopting this targeted approach of CRISPR-based enrichment could greatly reduce the required input and resources for achieving haplotype-resolved assemblies. To demonstrate the possibility, we chose the 4.5 megabase (Mb), notoriously difficult-to-assemble major histocompatibility complex (MHC) region, which contains many genes directly involved in immune responses to antigens and is associated with many complex human diseases[15–17]. The studies of the MHC region are hampered by its extremely high polymorphisms and strong linkage disequilibrium. Achieving haplotype-resolved assembly of the MHC region is vital for unbiased detection and characterization of genetic variants in the MHC region[18] for accurate genome inference[19] towards the eventual understanding of its molecular mechanisms in disease etiology.

In this work, by combining the CRISPR-based enrichment with 10x Genomics linked-read sequencing and the PacBio HiFi long-read platform (Fig. 1a), we were able to achieve targeted haplotype-resolved assembly of the MHC region from diploid human cells. We also applied our targeted approach to two challenging medically relevant genes and obtained haplotype-resolved assemblies. By comparing to the haplotype-resolved genome assembly from the same cell line, we showed that our targeted approach for obtaining haplotype-resolved assembly achieved comparable completeness and accuracy, enabling comprehensive detection of genetic variants of the targeted region. With the targeted assembled personal MHC haplotypes available as the reference, we showed that the quantifications of DNA methylation and gene expression of the MHC region are much more accurate, compared to the standard approach using the hg38 as the reference. Finally, we used the targeted assembled personal MHC haplotypes to investigate allele-specific transcriptional regulation of the MHC region.

## Results

### CRISPR-based targeted enrichment of the MHC region
Our studies employed the well-characterized and widely used human diploid cell line, GM12878 (also known as HG001). The first step of our approach is the isolation of the targeted megabase-size DNA region (the MHC region) from agarose-embedded, intact cells by combining CRISPR-based in-gel digestion with pulsed-field gel electrophoresis (Supplementary Fig. 1a). Four agarose plugs equal to a total of $4 \times 10^6$ cells were used for the isolation of the MHC region. SgRNAs were designed to target the non-polymorphic DNA sequences flanking the 4.7 Mb region known to include the entire MHC locus[20] (Fig. 1b). The performance of the designed sgRNAs was initially evaluated through in vitro cleavage assays with PCR products amplified from the targeted regions (Supplementary Fig. 1b). Most of the final enriched DNA molecules were still high molecular weight (HMW) with length longer than 50 kb (Supplementary Fig. 2), and are thus suitable for subsequent long-read sequencing.

To evaluate the enrichment efficiency, we performed qPCR analyses of three genomic loci within the targeted MHC region (chr6: 28903952-33268517), which revealed more than 40-fold enrichment compared to the no sgRNA controls; note that no enrichment was detected for the MHC-flanking region (Fig. 1b). Sequencing with the Illumina short-read platform showed that the whole targeted MHC region was successfully enriched (Fig. 1c). Notably, the coverage within the entire 2.3 Mb DNA fragment is relatively the same but quite different between two fragments (Fig. 1c), suggesting that the enrichment of the targeted region is dependent on the cleavage efficiency of corresponding sgRNAs. These results showed that, through the CRISPR/Cas9 cleavage followed by gel purification, we can successfully enrich the targeted MHC region.

### Targeted haplotype-resolved assembly of the MHC region
After the enrichment of the MHC region, we constructed 10x Genomics linked-read libraries with the HMW MHC molecules. The phased variants within the targeted MHC region were called (Supplementary Fig. 3a) and then compared to two benchmark datasets for GM12878 cells (the GIAB (v.4.2.1)[21] and the Illumina Platinum Genomes[22]). Most of the variants were identified with false negative rates (FNRs) lower than 11% (Table 1). The accuracy of phasing was high, with both switch error rates and Hamming error rates lower than 1% (Supplementary Table 1). We estimated the HLA types with HLA-VBSeq[23] (Supplementary Fig. 3a) for six major HLA genes, which have been found to be strongly associated with many human diseases[24,25]. Except for one of the HLA-C alleles, all of the other alleles reached an 8-digit resolution, which is better than a recent prediction of HLA alleles based on whole-genome assemblies generated using the Oxford Nanopore data[26] (Supplementary Table 2). We further assembled the two haplotypes using our 10x Genomics linked-read data from the targeted MHC region and generated 18 contigs for each haplotype. For both haplotypes, the NGA50 scaffold size was consistently over 0.513 Mb with the largest contig length over 1.93 Mb (Supplementary Table 3).

Even though 10x Genomics linked-read sequencing performed well for phased variant calling, we were only able to assemble a pseudo-haplotype of the targeted MHC region. We, therefore, conducted additional sequencing using the PacBio HiFi platform. While the current deployment of the PacBio HiFi platform for genome assemblies requires large amounts of starting materials (e.g., more than 25 μg of genomic DNA from diploid human cells, in order to generate a minimum of 80 Gb data required for whole-genome assemblies), we were able to generate a 12 kb HiFi library with at least 60× coverage of the targeted MHC region (Supplementary Fig. 3b) from 20 ng of enriched HMW DNA. After filtering the HiFi reads belonging to the targeted MHC region, we were able to generate a primary assembly and eventually a haplotype-resolved assembly of the targeted MHC region by using the PacBio HiFi data only (Supplementary Fig. 3c). This assembly result contains 6 contigs for haplotype 1 and 4 contigs for haplotype 2, and covers more than 96% of the targeted MHC region for both haplotypes (Supplementary Table 4).

We again estimated the HLA types for six major HLA genes. Even though all of them reached the 8-digit resolution, we noticed that the phasing information for the three HLA class I genes is incorrect (Supplementary Table 5). This assembly result generated from one single sequencing platform suggests that at least two platforms are needed to achieve high-quality haplotype-resolved assemblies when trio information is not available, which is consistent with other recent findings[12,18].

By the intersection of variants generated from both 10x Genomics linked-read data and HiFi reads, we were able to obtain a highly reliable set of phased heterozygous variants to separate the HiFi reads into two haplotype-partitioned read sets. Eventually, two MHC haplotypes were assembled separately from each haplotype-partitioned HiFi read set, together with untagged HiFi reads (Fig. 2a). The targeted assembled MHC haplotypes cover most of the MHC region of the hg38 reference (Table 2). We did observe several breakpoints in our assembly result (Supplementary Table 6), and they seem to be resulting from the absence of HiFi reads (Supplementary Fig. 4a, Supplementary Table 6) from biased amplification.

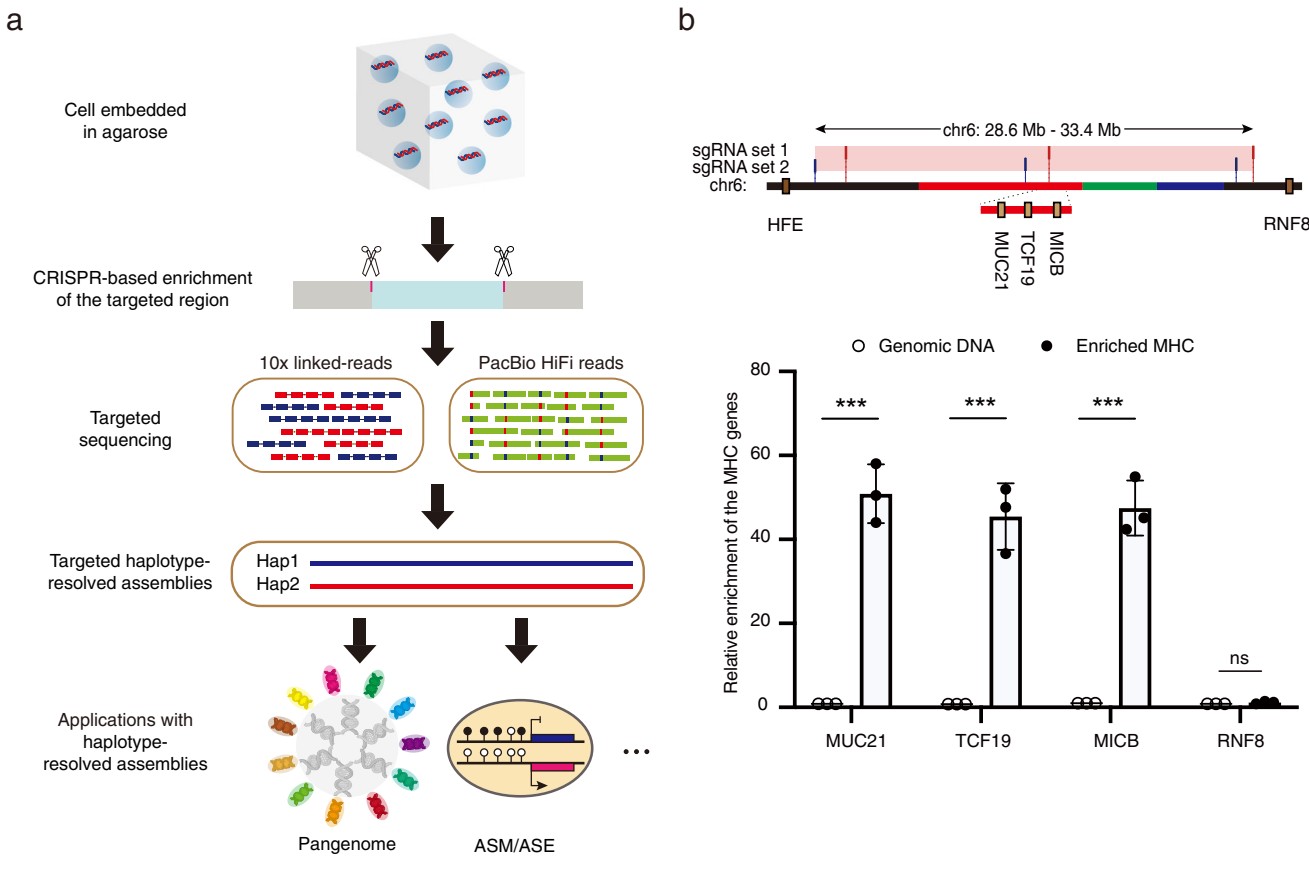

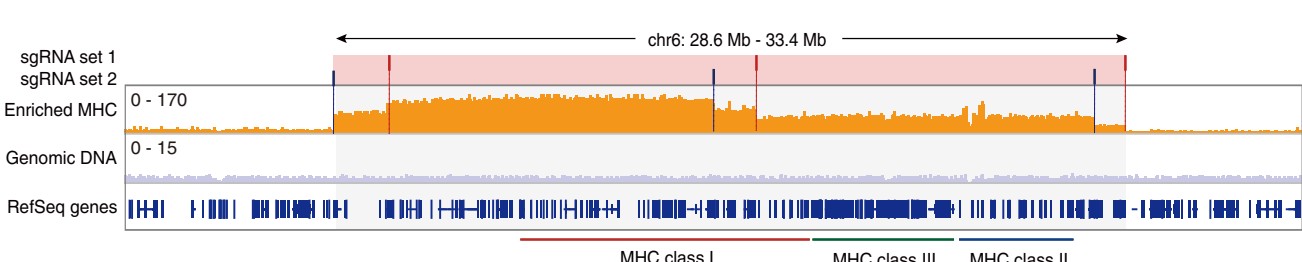

**Fig. 1 | CRISPR-based targeted enrichment of the MHC region. a** Schematics of CRISPR-based targeted haplotype-resolved assembly of the target region. Cells were embedded in agarose plugs and the targeted region was then enriched by combining CRISPR-based in-gel digestion with pulsed-field gel electrophoresis. Enriched HMW DNA molecules were subjected to both 10x Genomics linked-read and the PacBio HiFi long-read sequencing. A highly reliable set of phased heterozygous variants was called and used to separate the HiFi reads into two haplotype-partitioned read sets, from which haplotype-resolved assembly of the targeted region was constructed. The targeted assembled personal haplotypes can be further used in the downstream analyses. **b** QPCR analyses showed significant enrichment of the targeted MHC region. Top part: the positions of two sets of sgRNAs targeting the MHC region. The pink bar indicates the targeted MHC region. The red, green, and blue lines indicate the regions for the HLA class I, class III, and

class II genes, respectively. The dark yellow boxes represent genes used for qPCR analyses. Genes are listed in order based on their coordinates on the hg38 reference but not to scale. Bottom part: the relative enrichment was determined relative to that from the *HFE* gene (indicated in the top part) outside of the targeted MHC region, and then normalized to that from cells treated with no sgRNA. The relative enrichment for another gene outside of the targeted MHC region (*RNF8*) was also tested as the negative control. Data are represented as mean ± SEM from three independent experiments, *P* values: *MUC21 P* = 0.000248, *TCF19 P* = 0.000628, *MICB P* = 0.000261, *RNF8 P* = 0.7499 (****P* < 0.001, ns: not significant, Student's *t*-test, two-sided). Source data are provided as a Source Data file. **c** The coverage of the targeted MHC region (the gray area) based on sequencing with the Illumina short-read platform. The targeted sites for two sets of sgRNAs are indicated as red and blue bars, respectively.

We typed the six major *HLA* genes again, and all of them reached the 8-digit resolution (Fig. 2b). We noticed that the sequence of one *HLA-C* allele we identified is different from all the available *HLA-C* alleles reported in the IMGT/HLA database. Through Sanger sequencing, we confirmed that it is a novel *HLA-C* allele with one nucleotide difference from two best-matching IMGT/HLA alleles (*C\*01:02:01:02* and *C\*01:02:01:03*) (Fig. 2c). These results showed that with our

targeted haplotype-resolved assembly of the MHC region, the HLA alleles can be typed with high accuracy.

**Evaluation of targeted haplotype-resolved assembly of the MHC region**

To evaluate the accuracy of our assembly result, variants were called by aligning the phased contigs against the hg38 reference and then

**Table 1 | Genetic variants called from 10x Genomic linked-read data and the assembled haplotypes**

|  | 10x linked-read | HiFi only assembly | Targeted assembly | Garg et al.[8] |
|---|---|---|---|---|
| All variants | 28,938 | 29,721 | 30,100 | 30,001 |
| Phased variants | 27,244 | 29,721 | 30,100 | 30,001 |
| Heterozygous variants (GIAB) | 15,385 (2) | 8950 (8389) | 17,370 (1) | 17,384 (11) |
| Heterozygous variants (Illumina) | 14,931 (2) | 7904 (7671) | 15,616 (0) | 15,523 (8) |
| FNR (GIAB) | 0.10714 | 0.02901 | 0.02728 | 0.02475 |
| FNR (Illumina) | 0.03599 | 0.01439 | 0.01268 | 0.01679 |
| FDR (GIAB) | 0.23338 | 0.18423 | 0.19314 | 0.18654 |
| FDR (Illumina) | 0.27994 | 0.28221 | 0.29003 | 0.29063 |

The number of variants with different phasing information from the two benchmark datasets is indicated in the brackets. FNR: False negative rate, the percentage of true variants that are missed in the assembly; FDR: False discovery rate, the percentage of assembly-based variant calls that are not present in the benchmark.

compared to the benchmarks of the GIAB (v.4.2.1) and the Illumina Platinum Genomes. We identified 30,100 genetic variants within the targeted MHC region with FNRs lower than 2.8% (Table 1). Evaluation of variants using bed files from high-confidence regions showed that F1 scores (representing a harmonic mean of precision and recall) were higher than 0.89 for both benchmarks (Supplementary Table 7). All variants were phased with both switch error rates and Hamming error rates lower than 0.45% (Supplementary Table 1), confirming the high quality of phasing. Recently, a chromosome-scale high-quality haplotype-resolved assembly of GM12878[8] was reported using the PacBio HiFi and Hi-C data. Notably, our targeted haplotype-resolved assembly of the MHC region achieved a comparable consensus accuracy with Garg et al. (Table 1, Supplementary Tables 1 and 7).

In comparison, we also evaluated the haplotype-resolved assembly result generated from the PacBio HiFi data only. Compared to the benchmarks of the GIAB (v.4.2.1) and the Illumina Platinum Genomes, a total of 29,721 genetic variants were identified with FNRs lower than 3% (Table 1). While switch error rates were lower than 0.5% (Supplementary Table 1), a considerable number of heterozygous variants were called with inaccurate phasing information (Table 1) with Hamming error rates higher than 45% (Supplementary Table 1). This is consistent with the result that the phasing information for the three HLA class II genes is incorrect (Supplementary Table 5). In fact, this is the result of a single switch error in the middle of the targeted MHC region (Supplementary Fig. 3d). Thus, this suggests that, even though we were able to generate a haplotype-resolved assembly of the targeted MHC region with relatively good quality by using the PacBio HiFi data only, the completeness and phasing accuracy of the targeted assembly can still be improved by combining with another strand-specific sequencing technology, such as the 10x Genomics linked-reads.

To further evaluate the performance of our targeted approach against the approach for whole-genome assembly, we compared our targeted haplotype-resolved assembly using both 10x Genomics linked-read data and PacBio HiFi reads to Garg et al., and observed a high consistency across the whole targeted region (Fig. 2d, Supplementary Fig. 4b). Similar to Garg et al., we identified 19.3% and 29.0% new genetic variants compared to the GIAB (v.4.2.1) and the Illumina Platinum Genomes datasets, respectively (Table 1). Based on the manual inspection of some of these newly identified variants, most of which are located in highly polymorphic or repetitive regions, and confirmed their accuracies by identifying supporting high-confidence HiFi reads (Fig. 3a). In order to evaluate the assembly accuracy of complex regions with repetitive content which have traditionally been collapsed or excluded, we estimated the number of base pairs that are collapsed in the assembly results by analyzing raw HiFi read coverage

of the whole targeted MHC region. Compared to the whole-genome assembly result, our targeted approach yields similar or less collapsed repeats for both assembled haplotypes (Supplementary Fig. 4c, Supplementary Data 1). Further evaluation of the segmental duplication region with important genes *C4A/B*, *TNXA/B*, and *CYP21A2* showed that 20 kb of collapsed repeats were observed in this region of haplotype 1 from Garg et al., while there is no evidence of collapsed repeats from our targeted approach (Fig. 3b, Supplementary Data 1). Consistently, when comparing the assembly results to the bed files of this region from both benchmarks, F1 scores were higher than 0.83 for our targeted result, while they were lower than 0.6 in Garg et al. (Supplementary Data 2). All these results highlight that our targeted approach can achieve a high level of accuracy, comparable to the whole-genome assembly result.

With the targeted haplotype-resolved assembly available, we were able to identify large insertions and deletions, which are difficult to find using short-read sequencing. For example, we identified a 2889 bp deletion upstream of the *HCP5* gene (Fig. 3c) and a region with two insertions (344 and 787 bp) (Supplementary Fig. 4d), both of which are supported by the presence of HiFi long reads and observed in Garg et al. It has been reported that the assembly-based approach enables more accurate and comprehensive characterization of genetic variants[1,18], and by comparing to the haplotype-resolved genome assembly, our results showed that the targeted approach can achieve comparable completeness and accuracy on the detection of genetic variants in the targeted region.

### Accurate functional genomics analyses with the targeted haplotype-resolved assembly

It has been reported that the most polymorphic parts of the MHC locus are located at regions around three HLA class I genes (*HLA-A*, *HLA-B*, and *HLA-C*) and three HLA class II genes (*HLA-DR*, *HLA-DQ*, and *HLA-DP*)[24,27]. Consistently, our targeted haplotype-resolved assembly showed that the major peaks of variants are in genomic regions around these six classical *HLA* genes (Supplementary Fig. 5a). While a high level of polymorphisms within the MHC region can be beneficial for a population facing an environment with various pathogens[28], short sequencing reads carrying many polymorphisms and structural variants will not be aligned properly to a reference genome, leading towards ambiguous and even inaccurate results[19,29].

It has been proposed that the use of personal genome as the reference is a solution to deal with alignment-related artifacts for both gene expression[30,31] and DNA methylation analyses[32]. In light of this, we replaced the hg38 reference with personal genome references, in which two targeted assembled personal MHC haplotypes were combined separately with the non-MHC region of the hg38. Short-read sequencing data is then aligned to each personal genome reference individually to limit the effects of genetic variation on sequence alignment (Supplementary Fig. 5b). Consistent with previous findings[30,31], using the hg38 as a reference for the sequence alignment of the RNA-Seq data generated from GM12878 cells, the number of sequencing reads mapped to the exons harboring many variants (regions highlighted with the gray shadow in Supplementary Fig. 5c) were much lower compared to the other exons from the same *HLA-B* gene. However, the effect of poor alignment was dramatically improved when we used personal genomes as the reference (Supplementary Fig. 5c). This highlights the importance of utilizing the targeted haplotype-resolved assembly for accurate quantification of gene expression for genes with high polymorphisms.

This becomes more complicated in the study of DNA methylation. It is well-known that methylated cytosines are prone to spontaneous deamination, resulting in the most common dinucleotide variant (CG → TG) in mammalian genomes[33,34]. However, the gold standard to quantify the level of DNA methylation at each CpG site after bisulfite-conversion is to calculate the number of reads sequenced with CG

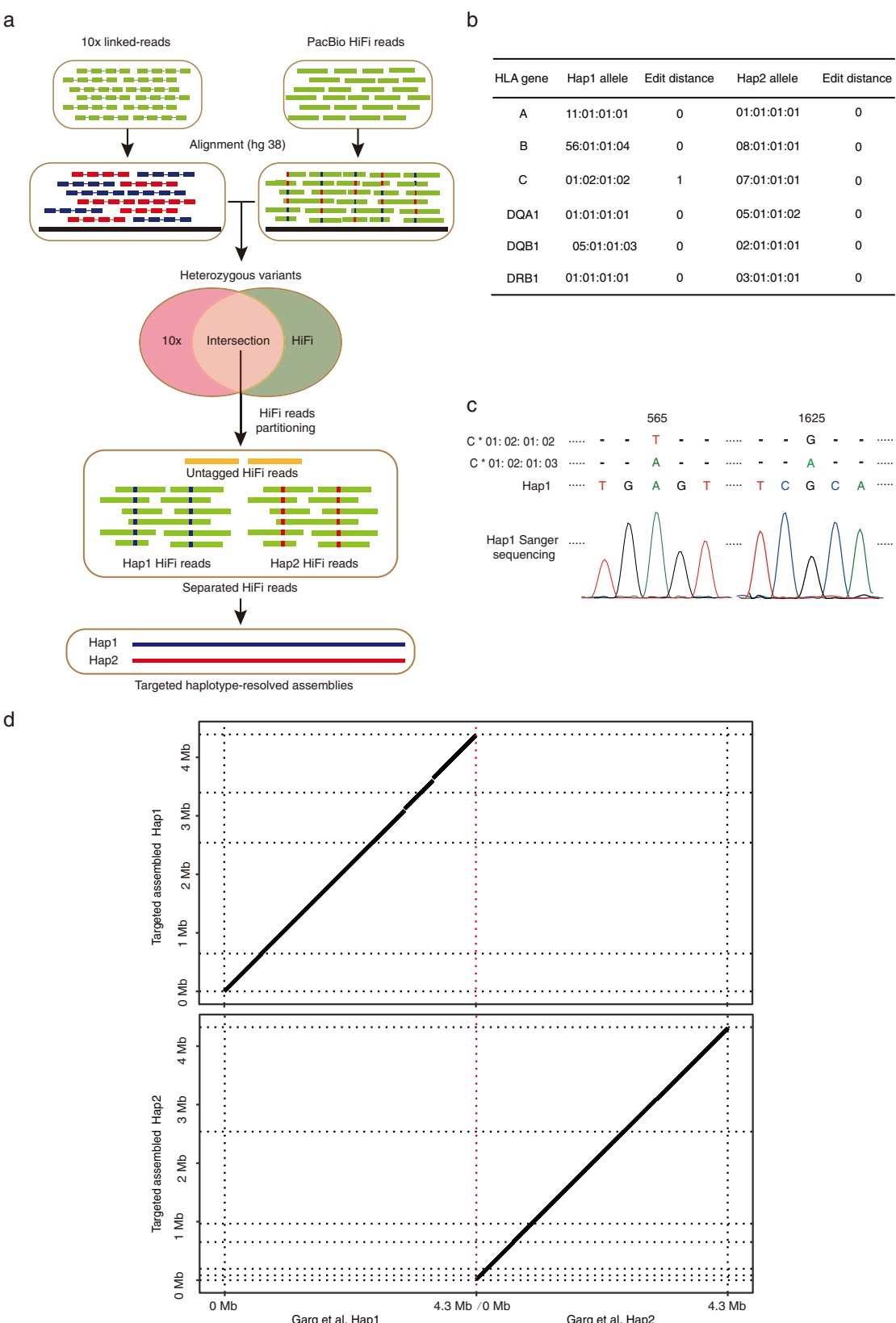

**Fig. 2 | Targeted haplotype-resolved assembly of the MHC region and HLA typing. a** Schematics of targeted haplotype-resolved assembly of the MHC region with 10x linked-read data and PacBio HiFi reads. **b** HLA typing of six classical *HLA* genes using the targeted haplotype-resolved assembly. Edit distance: the number of different nucleotides between our assemblies and the reported HLA alleles from the IMGT/HLA database. **c** A novel *HLA-C* allele in haplotype 1 is confirmed by Sanger sequencing. The nucleotides different from the best-matching IMGT/HLA alleles are shown. The positions of the different nucleotides in the *HLA-C* gene are indicated. **d** The comparison between our targeted assembly and the MHC region from genome assembly reported previously[8] for each haplotype. The *Y*-axis indicates the coordinate of our targeted assembly, and the *X*-axis indicates the coordinate of Garg et al.

**Table 2 | Statistics of two targeted assembled MHC haplotypes**

| | Haplotype 1 | Haplotype 2 |
|---|---|---|
| **Aligned to the hg38 reference** | | |
| Fraction of the targeted region (%) | 98.368 | 95.847 |
| Total aligned length (bp) | 4,311,531 | 4,194,872 |
| NGA50 (bp) | 636,872 | 456,634 |
| LGA50 | 3 | 4 |
| **Without the reference** | | |
| Number of contigs | 4 | 6 |
| Largest contig length (bp) | 1,895,202 | 1,784,832 |
| Total length (bp) | 4,391,445 | 4,323,583 |

NGA50: minimum contig alignment length needed to cover 50% of the hg38 reference.
LGA50: the number of contigs when reached NGA50.

(indicating bisulfite-protected by the methyl group) divided by the number of reads sequenced with TG (indicating bisulfite-converted without the methyl group). In this situation, C/T variants on CpG sites will inevitably lead to erroneous methylation quantifications. For example, a specific TG-to-CG variant was observed in the targeted assembled haplotype 2 positioned downstream of the *HLA-B* gene (the left gray box in Fig. 4a). For this position, if the sequencing reads were aligned to the hg38 reference, they will be missed from the evaluation of DNA methylation. Alternatively, another CG-to-TG variant observed in haplotype 2 (the right gray box in Fig. 4a) will produce unmethylated calls without inducing any alignment mismatches and result in an excess of 0% methylated 'sites' at this position where there is actually no CpG site in the haplotype 2.

We evaluated the CpG sites among two targeted assembled MHC haplotypes and the hg38 reference and found that 2.38% (1355/57,010) of CpGs are specific in haplotype 1 and 2.14% (1191/55,659) are specific in haplotype 2. However, 3.91% (2291/58,570) of CpGs in the hg38 reference are not present in either haplotype (Supplementary Fig. 6a). When characterizing the distribution of these CpG variants (haplotype-specific gains and losses of CpGs relative to the hg38) spanning the targeted MHC region, we observed that the major peaks are present in genomic regions around the six classical *HLA* genes (Fig. 4b), similar to the density plot generated from genetic variants (Supplementary Fig. 5a). The problematic effect resulting from CpG variants was obvious when we performed the DNA methylation analyses separately for haplotype-specific or shared CpGs. As expected, the extreme discrepancy was observed when computing the methylation level for haplotype-specific CpGs using the corresponding targeted assembled personal MHC haplotype as the reference, while this difference was not observed for shared CpGs (Fig. 4c). With the presence of the targeted haplotype-resolved assembly of the MHC region, this bias can be easily addressed by aligning the bisulfite sequencing data to personal genomes as the reference (Supplementary Fig. 5b), similar to what has been shown previously for analyzing methylation data on two highly divergent mouse strains[32].

Bias for methylation quantification is also observed with DNA methylation arrays, reflecting that the high level of polymorphisms spanning the MHC region will affect the hybridization behavior of SNP-associated probes[35]. We observed a correlation coefficient ($R^2$) of 0.96 for DNA methylation between the Illumina methylation EPIC beadchip and bisulfite sequencing data for CpGs without SNPs in the array probes (Supplementary Fig. 6b), while the correlation coefficient decreased to 0.9 for the probes harboring SNPs (Supplementary Fig. 6c). These results collectively highlight the utility of using the targeted assembled personal haplotypes as the reference for accurate quantifications of DNA methylation and gene expression data, especially for highly polymorphic regions such as the MHC.

## Allele-specific analyses with the targeted haplotype-resolved assembly

With the targeted assembled MHC haplotypes available, we further characterized allele-specific methylation and expression for diploid GM12878 cells. We first analyzed the allele-specific expression (ASE) using two personal genomes as the reference and identified that seven genes within the targeted MHC region exhibit allele-specific expression (adjusted *P* value ≤ 0.05) (Supplementary Fig. 7a and Supplementary Data 3). For the expression of the *HLA-DPA1* gene, which had a 1.1-fold difference between the two haplotypes, we observed consistently higher expression for all of the haplotype 2 exons of *DPA1* (Fig. 4d). This ASE was validated using pyrosequencing (Fig. 4e), allele-specific qRT-PCR (*P* value = 0.0074, paired Student's *t*-test, two-sided) (Supplementary Fig. 7b) and Sanger sequencing of amplified PCR clones (*P* value = 0.0068, Fisher's exact test) (Supplementary Fig. 7c), and all of them showed that the haplotype 2 of *DPA1* gene is significantly overexpressed compared to the haplotype 1 in GM12878 cells. The fact that the expression of the *DPA1* gene in the haplotype 2 (*DPA1\*01:03:01:02*) is higher than that in the haplotype 1 (*DPA1\*02:01:01:02*) in GM12878 cells is consistent with a recent finding that the expression of the *DPA1\*01* allele is significantly higher than that of the *DPA1\*02* allele in the population[36].

We then analyzed the allele-specific methylation (ASM) for the targeted MHC region and found 211 differentially methylated regions (DMRs) between the two haplotypes (Supplementary Data 4). We included haplotype-specific CpGs in the methylation analysis, as this will increase the accuracy and power in the differential analysis[32] (Supplementary Fig. 7d). We noticed that, in the promoter region of the *HLA-DRA1* gene, there was a differentially methylated region (termed DMR-DPA1 hereafter), in which the haplotype 1 was hypermethylated (Fig. 4d). DMR-DPA1 contains one haplotype 1-specific CpG site and several shared CpGs, and the allele-specific methylation of DMR-DPA1 was validated using bisulfite Sanger sequencing (Fig. 4f).

To examine whether the ASM of DMR-DPA1 exerts any regulatory impacts on the ASE of the *DPA1* gene, we performed a cell-based dual luciferase reporter assay with HEK293T cells and detected a significant increase of gene expression when unmethylated DMR-DPA1 is present. In contrast, this upregulation of gene expression is abolished when DMR-DPA1 on the luciferase reporter was methylated in vitro (Fig. 4g). This suggests that the ASE of the *HLA-DPA1* gene can be negatively regulated through ASM on the promoter region of *HLA-DPA1*. These results illustrate that the targeted assembled personal haplotypes enable allele-specific functional genomics analyses in diploid human cells, and can contribute to our understanding of allele-specific transcriptional regulation of targeted genomic regions.

## Targeted assembly of challenging medically relevant genes

Recently, the GIAB Consortium showed that they resolved and curated variation benchmarks for 273 out of 395 challenging medically relevant autosomal genes from a haplotype-resolved whole-genome assembly[2]. To extend the application of our targeted approach, we applied our method to one GIAB-resolved gene (*RHCE*) and one GIAB-unresolved gene (*CR1*). The gene *RHCE* is part of the Rh blood group antigens[37] with SVs and complex variants. *CR1* is a gene implicated in Alzheimer's disease[38] and contains an 18.5 kb homozygous deletion relative to the hg38 in HG002 cells. These two gene regions were isolated from GM12878 cells either alone or together by using CRISPR/Cas9 cleavage followed by gel purification (Supplementary Fig. 8a, b). Both qPCR analyses (Supplementary Fig. 8c, d) and sequencing with the Illumina short-read platform (Supplementary Fig. 8e) showed that both targeted regions were successfully enriched.

We next used either the PacBio HiFi reads only or combined them with the 10x Genomics linked reads to assemble two targeted regions. Other than the HiFi-only assembly of the *CR1* gene, both assembly strategies generated haplotype-resolved assemblies of the two

a

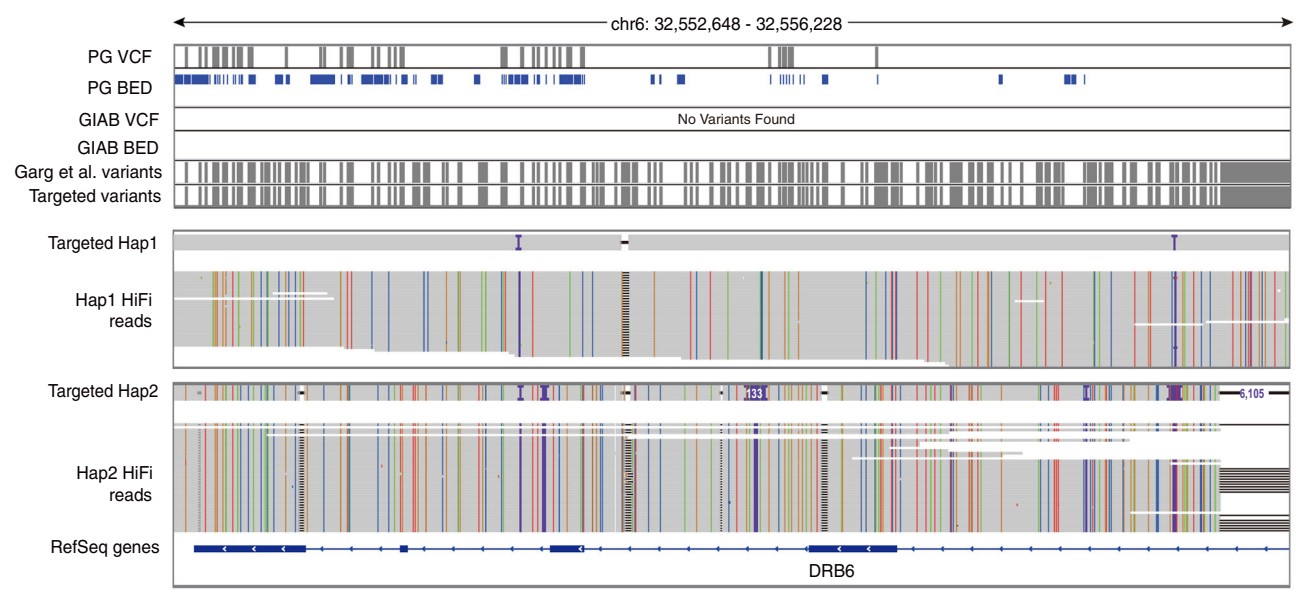

b

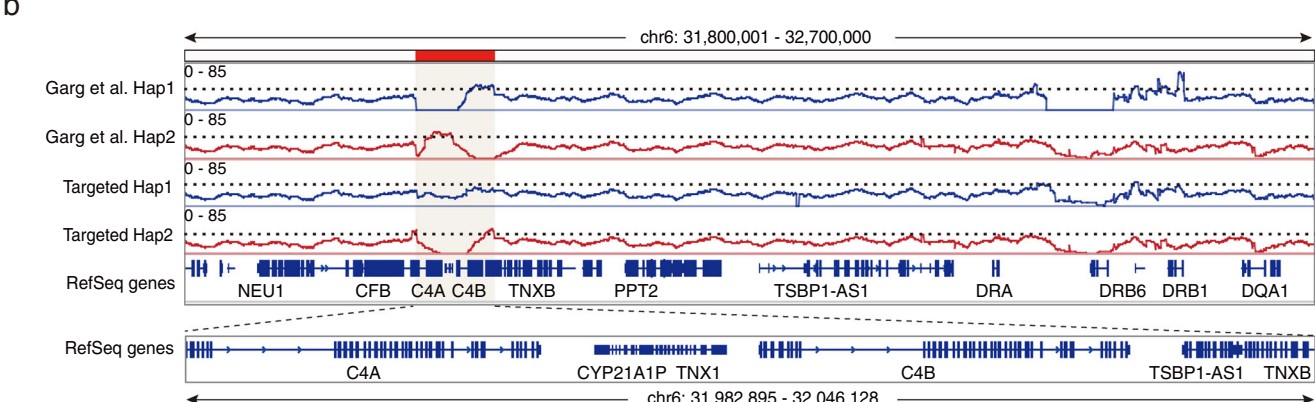

c

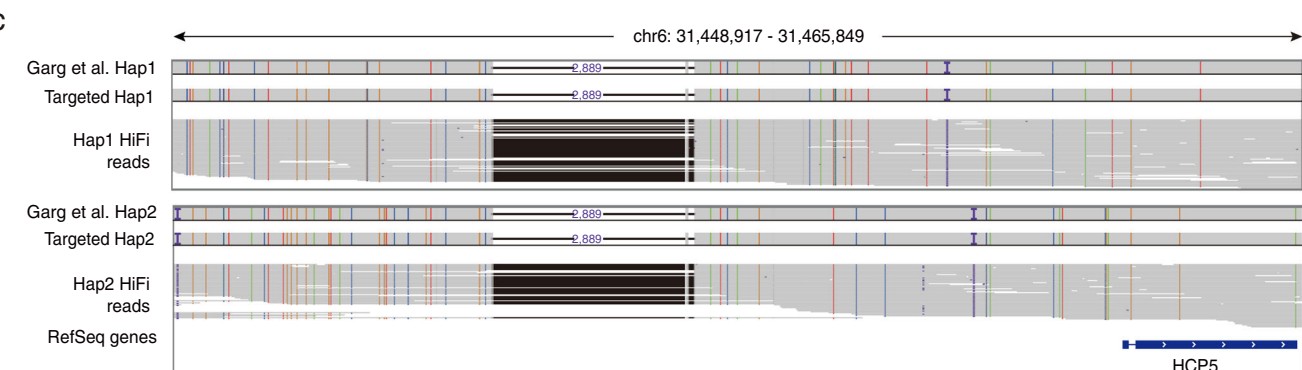

**Fig. 3 | Accurate genetic variant calling using the targeted haplotype-resolved assembly. a** An example of a region with newly identified variants. High-confidence HiFi reads with new variants are illustrated below. The colorful lines and dots positioned in the targeted assembly result of the haplotype 1 and HiFi reads indicate four types of nucleotides different from the hg38 reference. The purple character "I" and dots indicate small insertions (<50 bp) compared to the hg38 reference. Black dots indicate small deletions (<50 bp) compared to the hg38 reference. PG: the Illumina Platinum Genomes. **b** Collapsed region in the assemblies. Blue and red lines represent the coverage of downloaded PacBio HiFi reads of GM12878 cells from the GIAB aligned to the haplotype 1 or haplotype 2 of different assemblies. The dotted lines indicate the threshold of expected coverage (mean + at least three standard deviations) of chromosome 6 excluding the targeted MHC region for the collapsed region. The red bar indicates the collapsed region identified in Supplementary Data 1. **c** A 2889 bp deletion upstream of the *HCP5* gene is illustrated and supported by HiFi reads for both haplotypes.

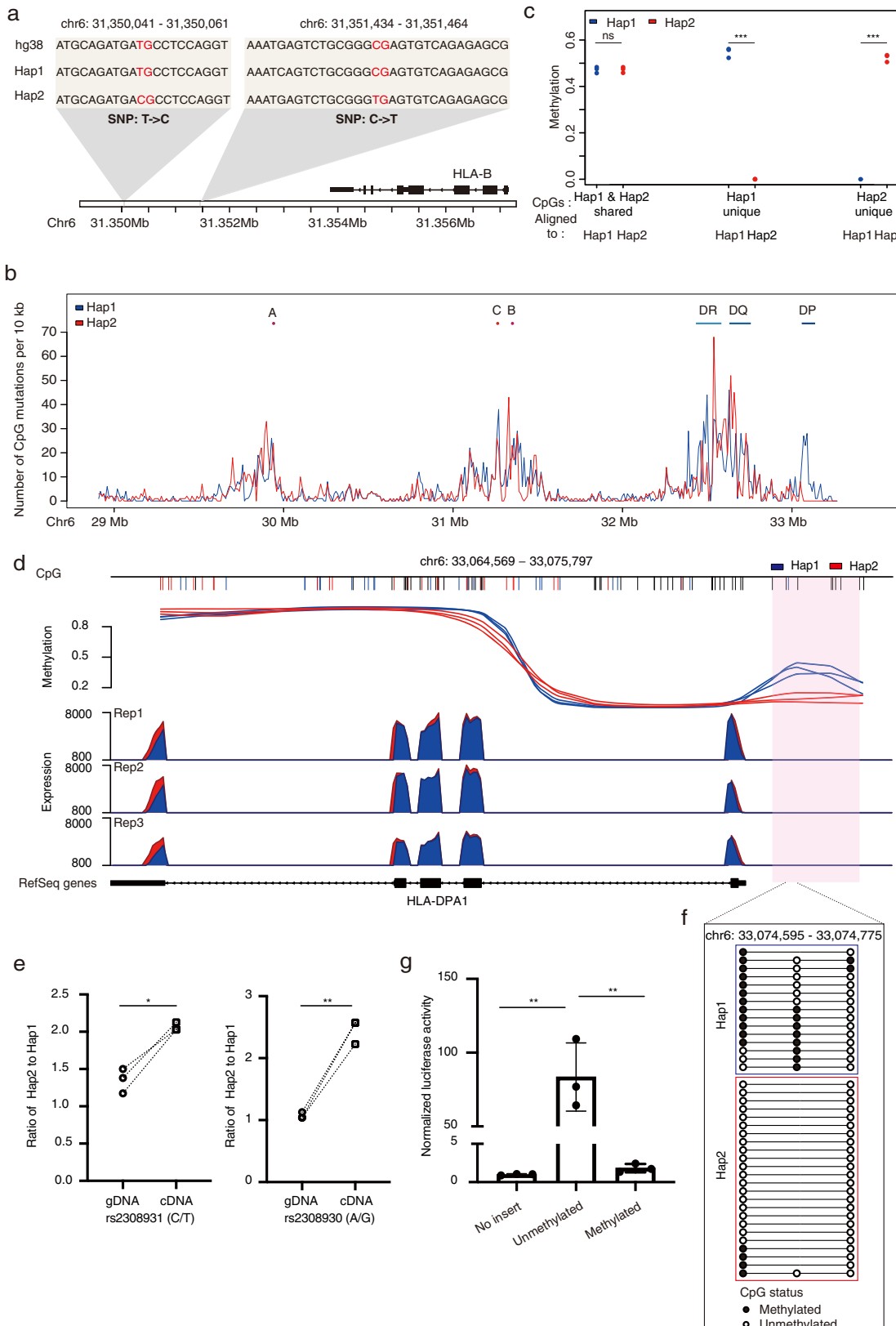

targeted genes with only 1 contig for each haplotype (Supplementary Table 8). We evaluated the accuracy of called variants from the assemblies of the entire targeted regions by comparing them to the two benchmarks and observed that, for both genes, both assembly strategies can identify most genetic variants with FNR lower than 7% (*RHCE*) and 3% (*CR1*) (Supplementary Data 5). Since challenging

medically relevant regions of *RHCE* and *CR1* genes were not well-resolved in both benchmarks, we identified many new variants, resulting in a high level of FDRs (more than 40%) (Supplementary Data 5). Comparing the bed files from two benchmarks also showed high accuracy of called variants (Supplementary Data 6). The switch errors and Hamming errors were low for both assemblies, except that

**Fig. 4 | Accurate and allele-specific functional genomics analyses using the targeted haplotype-resolved assembly. a** Examples of C/T variants on CpG sites. **b** The CpG variant density plot throughout the targeted MHC region. The positions of three MHC class I genes and three MHC class II genes are indicated at the top. **c** The effect of CpG variants on DNA methylation analysis. Data are represented from three independent experiments, *P* values: shared *P* = 0.851, Hap1 unique *P* = 0.000496, Hap2 unique *P* = 0.000318 (***P* < 0.001, ns: not significant, paired Student's *t*-test, two-sided). **d** Allele-specific transcriptional regulation of the *HLA-DPA1* gene. Top part: allele-specific methylation in the promoter region of the *HLA-DPA1* gene. The differentially methylated region (DMR-DPA1) between two haplotypes is indicated as a pink bar. The shared CpGs between two haplotypes are shown as black short lines, while haplotype-specific CpGs are indicated as blue or red short lines. The *Y*-axis indicates the DNA methylation level for each haplotype, separately. Three replicates are shown. Bottom part: allele-specific expression of the *HLA-DPA1* gene. The expression level for haplotype 1 and 2 is indicated as blue and red, respectively. Three replicates are shown. **e** Pyrosequencing of two polymorphic positions within the exons of the *HLA-DPA1* gene of GM12878 cells. *Y*-axis indicates the ratio of the targeted SNPs between the two haplotypes. Data are represented from three independent experiments, *P* values: rs2308931 (C/T) *P* = 0.0177, rs2308930 (A/G) *P* = 0.0056 (*0.01 < *P* < 0.05, **0.001 < *P* < 0.01, paired Student's *t*-test, two-sided). **f** Bisulfite Sanger sequencing of PCR clones for part of the DMR-DPA1 region, containing one haplotype 1-specific CpG and two haplotype-shared CpGs. Each line represents one read where black or white circles illustrate methylated or unmethylated CpGs, respectively. **g** DMR-DPA1 exerted methylation-dependent promoter activity on gene expression. Dual-luciferase reporter assays were performed. Data are represented as mean ± SEM from three independent experiments, *P* values: Unmethylated/No insert *P* = 0.0035, Methylated/Unmethylated *P* = 0.0036 (**0.001 < *P* < 0.01, Student's *t*-test, two-sided). Source data are provided as a Source Data file.

for the HiFi-only assembly of the *CR1* gene, the Hamming errors were high (Supplementary Data 5).

For both *RHCE* and *CR1* genes, the assembly results using both 10x Genomics linked-read data and PacBio HiFi reads were supported by the presence of PacBio HiFi reads across the targeted region (Supplementary Fig. 9), with some breaks in the alignments of the assemblies to the hg38 reference (Supplementary Fig. 9). Recently, a new version of minimap2 (v2.24) was released, which improves alignment strategies around long poorly aligned regions. In order to distinguish whether these breaks are due to breaks in the assemblies or in the alignments, the alignment results were further compared with different versions of minimap2 (Supplementary Fig. 10). While no change was observed for the *RHCE* gene, two large deletions (17,095 and 18,555 bp) were identified in the haplotype 1 of *CR1* gene using the latest version of minimap2 (v2.24) (Supplementary Fig. 10). All these results demonstrated that similar to haplotype-resolved whole-genome assemblies, our targeted approach can be used to resolve and characterize genetic variants in other genomic complex regions.

## Discussion

The recent advances in constructing high-quality haplotype-resolved genome assemblies have made it possible to comprehensively discover genetic variants at the chromosome scale. However, this genome-wide approach is difficult to be applied in a large group of individuals or even among diverse populations to fully characterize genetic variants in order to reveal their potential functional relevance. In this study, using the clinically important MHC region and two challenging medically relevant genes as examples, by combining the CRISPR-based enrichment with 10x Genomics linked-read sequencing and the PacBio HiFi platform, we successfully assembled the targeted haplotypes without parent–child trio information. By comparing to the haplotype-resolved genome assembly, our targeted approach for obtaining haplotype-resolved assembly achieved comparable completeness and accuracy with reduced computing complexity, sequencing cost, as well as the amount of starting materials (Supplementary Tables 9 and 10).

Even though genome-wide association studies (GWAS) have greatly advanced our understanding of the human genome, it is still challenging to pinpoint causal variants involved in disease etiology for genetic regions with high polymorphisms and strong linkage disequilibrium (e.g., the MHC region). It has been suggested that haplotype-based approaches may overcome these challenges toward more accurate genome inference[19]. One example of this is the haplotypes of *HLA-DRB1-HLA-DQA1-HLA-DQB1* greatly increased the risk of type 1 diabetes[39]. Another example is the identification of different alleles of the complement component 4 (*C4*) genes within the MHC region for the development of schizophrenia[40]. Even though we only demonstrated our targeted approach using GM12878 cells, it has been shown that a CRISPR-based strategy for the enrichment of HMW DNA

molecules can be similarly applied to primary tissues, including peripheral blood mononuclear cells[11] and breast tumors[13]. Considering the MHC region is one of the most difficult regions to be assembled in the human genome[18], our targeted approach can (in theory) be adopted for haplotype-based population genetic studies to analyze any other complex genomic regions of interest like the MHC.

An issue that may complicate studies adopting our targeted approach is that the enrichment strategy of designing sgRNAs is dependent on the understanding of the targeted region, as well as the sequences flanking it. If the targeted region is duplicated, inverted, or deleted, this would change the molecular weight of enriched DNA molecules, making it difficult to be recovered from the pulsed-field gel. Moreover, even though generating targeted haplotype-resolved assembly from the PacBio HiFi reads with Hifiasm is independent of the reference genome, the reference sequence is still needed in order to filter the HiFi reads aligned to the targeted region.

In addition to genetic variants, functional analyses of other complex genomic regions can also benefit greatly from our unbiased approach to constructing haplotype-resolved assemblies of targeted regions. For regions with high polymorphisms, such as the MHC, the conventional strategy of mapping short-read sequencing data to a single reference genome (e.g., hg19 or hg38) is known to yield biased alignments, leading towards inaccurate quantifications[41,42]. This has prompted recent attempts to replace a single reference genome with the computationally inferred personal genotypes as the reference, all of which showed improved accuracies of quantifications compared to the standard approach[36,43–46]. However, these computational strategies highly depend on the completeness of our understanding of diverse human genomes, which is still far from saturation, even for the widely studied classical *HLA* genes[47]. Here, using the targeted assembled personal MHC haplotypes as an example, we not only showed that the results of functional genomics analyses of the MHC region are much more accurate, but also revealed allele-specific transcriptional regulation of the *HLA-DPA1* gene through the luciferase assay. As the luciferase assay was performed in the HEK293T cells, it should be noted that further experiments are needed to confirm this pattern is biological and holds in other cell types as well.

With that being said, the identification of ASE and ASM events accurately is quite challenging to do with short-read sequencing data. Even though our approach can limit the effects of alignment bias caused by genetic variations, other challenges, such as sequencing reads aligned to multiple genome locations or to both personal genome references, remain to be resolved. Nevertheless, it can be safely assumed that other functional genomics analyses of other challenging regions like the MHC would benefit from our approach of using the haplotype-resolved assembly of targeted regions as the personal reference. This integrated analysis of genetic and epigenetic data may pave a new way for comprehensively elucidating the molecular mechanisms in disease etiology.

## Methods

### Preparation of agarose plugs

Preparation of megabase-sized MHC DNA molecules or two challenging medically relevant autosomal genes from GM12878 cells was based on a method described previously[48] with modifications. Briefly, harvested GM12878 cells were washed three times in ice-cold PBS, resuspended to a final concentration of ~$2 \times 10^7$ cells/ml in L buffer (0.1 M EDTA (pH 8.0); 0.01 M Tris−Cl (pH 7.6); 0.02 M NaCl), and incubated at 42 °C for 5 min. 1.2% low-melting-point (LMP) agarose was melted at 70 °C followed by incubation at 42 °C for 5 min and mixed 1:1 with the cell suspension. The mixture was immediately dispensed into a plug mold (1 ml of the cell−agarose mixture can fill 10 wells in the mold) and plugs were incubated at 4 °C until solidified. Solid agarose plugs were transferred into 3 volumes of L buffer containing 0.5 mg/ml proteinase K and 1% (w/v) Sarkosyl and incubated at 50 °C for 3 h. The original digestion buffer was replaced with 2 volumes of fresh digestion buffer and incubation was continued for another 12–16 h at 50 °C. The plugs were rinsed three times in 50 volumes of TE buffer (10 mM Tris, pH 8, 1 mM EDTA) over a period of 3 h, and then incubated with 2 volumes of TE containing 40 µg/ml PMSF for 1 h at room temperature followed by another incubation with the same buffer at 50 °C for 30 min. Next, plugs were washed three times with TE buffer over a period of 3 h and used for Cas9 digestion or stored in TE buffer at 4 °C for future usage.

### Generation of sgRNAs

Two sets of sgRNAs with 20-bp sequences (20-mers) from non-polymorphic genomic regions flanking the targeted MHC region (set1: chr6: 28.90–33.46 Mb; set2: chr6: 28.58–33.27 Mb) (Fig. 1b), as well as sgRNAs targeting *RHCE* (chr1: 25.40–25.50 Mb) and *CR1* (chr1: 207.47–207.79 Mb) genes, were designed using CRISPRdirect (https://crispr.dbcls.jp) (Supplementary Data 7). The sgRNAs were in vitro transcribed using the EnGen sgRNA kit (NEB), and purified with the RNA Clean & Concentrator™-25 kit (Zymo Research). The purified sgRNAs were quantified by the NanoDrop (Thermo Fisher Scientific).

### CRISPR-based digestion of the targeted region

Agarose plugs containing GM12878 cells were in vitro digested with the *S. pyogenes* Cas9 Nuclease (NEB) based on a method described previously[49] with modifications. Briefly, sgRNAs and Cas9 enzyme were pre-assembled prior to the digestion by mixing 4 pmol of Cas9 enzyme with 150 ng of sgRNAs, 6 µl 10× Cas9 buffer, 1 U/µl RNasin, and nuclease-free water to a final volume of 60 µl with incubation at 37 °C for 15 min. The cleavage efficiency of each pre-assembled Cas9-sgRNA complex was evaluated in a 30 µl reaction volume that contains 30 nM of total sgRNA, 30 nM of Cas9 enzyme, and 3 nM of PCR amplified DNA fragments containing the Cas9 digestion site. Agarose plugs were washed three times in 10 mM Tris–HCl (pH 8.0) followed by incubation in 1× Cas9 buffer for 2 h. Four agarose plugs were used for the MHC isolation (~20 µg of genomic DNA), and two plugs for the medically relevant genes (*RHCE* and *CR1*) (~10 µg of genomic DNA). Plugs were then divided into two equal volumes of 40 µl and each was digested with one set of 60 µl pre-assembled Cas9-sgRNA mixes at 37 °C for 2 h. Reactions were terminated by the replacement of the reaction mix with L buffer containing 0.5 mg/ml proteinase K and 1% (w/v) Sarkosyl followed by at least 1 h incubation with gentle shaking at 4 °C.

### Targeted enrichment of the MHC region with pulsed-field gel electrophoresis (PFGE)

After Cas9 digestion, agarose plugs, together with *H. wingei* CHEF DNA Size Markers (Bio-Rad), were mounted directly onto the bottom edge of the gel comb and incorporated in 180 ml of 0.8% megabase

resolution gel prepared in 1× TAE (Bio-Rad). All PFGE was performed on a Bio-Rad CHEF-DR III system at a 106° angle and 3 V/cm for 48 h with a fixed switch time of 500 s. Gels were post-stained using 3× Gel-Green Stain in 0.1 M NaCl for 30 min at room temperature. The bands corresponding to ~2.3 Mb were cut out of the gel and placed into 4% LMP agarose in 1× TAE and separated again using PFGE at a 106° angle and 3 V/cm for 17 h with a fixed switch time of 500 s. The LMP gels were post-stained using 3× Gel-Green Stain in 0.1 M NaCl for 30 min at room temperature and the bands containing HMW DNA were cut from LMP gels for recovery.

### Targeted enrichment of the *RHCE* and *CR1* regions with PFGE

After Cas9 digestion, agarose plugs, together with lambda DNA (NEB) as markers, were mounted directly onto the bottom edge of the gel comb and incorporated in 180 ml of 1% gel prepared in 0.5× TBE (Sangon). All PFGE was performed on a Bio-Rad CHEF-DR III system at an auto pattern with 20–500 kb separation. Gels were post-stained using 3× Gel-Green Stain in 0.1 M NaCl for 30 min at room temperature. The bands corresponding to ~100 kb (*RHCE*) and ~300 kb (*CR1*) were cut out of the gel and placed into 4% LMP agarose in 0.5× TBE and separated again using PFGE at the auto pattern with 20–500 kb for 4 h. The LMP gels were post-stained using 3× Gel-Green Stain in 0.1 M NaCl for 30 min at room temperature and the bands containing HMW DNA were cut from LMP gels for recovery.

### Recovery of the targeted HMW DNA molecules

The recovered LMP gels were melted in a heat block at 70 °C for 10 min followed by incubation at 42 °C for 5 min. 4 U of agarase (Thermo Fisher Scientific) was added to 100 mg (~100 µl) of molten 4% LMP agarose, gently mixed, and incubated at 42 °C for 45 min to digest agarose into an HMW DNA solution. Meanwhile, 10 ml of TE buffer was placed into a 6 cm Petri dish per DNA sample, and 0.1 µm dialysis membrane (Millipore) was floated on the surface of the TE buffer for 15 min to hydrate the membrane. The HMW DNA solution was added as a single drop onto the center of the dialysis membrane with a wide-bore tip and dialyzed for 50 min at room temperature. The dialyzed DNA was then transferred to a 1.5 ml tube with a wide-bore tip, and quantified by the Qubit DNA high-sensitive assay (Thermo Fisher Scientific). Quantitative real-time polymerase chain reaction (qPCR) was performed to determine whether the targeted region was enriched at multiple positions with primer sequences listed in Supplementary Data 7.

### Library construction and sequencing

Genomic DNA of GM12878 cells was extracted using DNeasy Blood & Tissue Kit (Qiagen). Next-generation sequencing libraries with short-reads from genomic or the targeted enriched DNA were constructed using the Ultra II DNA Library Prep Kit for Illumina (NEB) and sequenced on the Illumina Nova Novaseq 6000 platform according to the manufacturer's instructions.

Sequencing libraries for the 10x Genomics platform were constructed using the Chromium Genome Library & Gel Bead Kit v2 (10x Genomics) with modifications. Briefly, in order to reduce barcode collisions, the targeted enriched HMW DNA molecules were 1:1 mixed with the lambda DNA (NEB), and a total of 200 pg of mixed DNA was used for droplet generation. 30 µl of generated droplets was aliquoted for amplification and subsequent library construction according to the manufacturer's instructions. The library was sequenced on the Illumina Hiseq X Ten platform according to the manufacturer's instructions.

HiFi SMRTbell library was constructed according to the Ultra-Low DNA Input Workflow (PacBio) with modifications. Briefly, 20 ng of the targeted enriched HMW DNA was purified with magnetic beads (ProNex, Promega, Madison, WI, USA) and fragmented with g-Tubes (Covaris). The constructed library was size-selected to 8–17 kb using

the BluePippin Size-Selection System. The final library with an average of 12 kb was sequenced on the Sequel II System (PacBio).

The bisulfite-sequencing libraries were prepared based on a method described previously[50] with modifications, by using EZ DNA Methylation-Gold Kit (NEB) and NEBNext® Ultra™ II DNA Library Prep Kit (NEB) from 10 ng of the targeted enriched HMW MHC DNA molecules, and sequenced on the Illumina Hiseq X Ten platform. For Illumina methylation EPIC beadchip, 500 ng of genomic DNA from GM12878 cells was used and the experiment was performed by Shanghai Genergy Co. Ltd (Shanghai, China) according to the manufacturer's instructions.

For strand-specific RNA-Seq library construction, total RNA was first extracted from GM12878 cells using Trizol (Thermo Fisher Scientific), and 1 μg of RNA was used to construct the sequencing library with the NEBNext Ultra Directional RNA Library Prep Kit (NEB). The constructed libraries were sequenced on the Illumina Hiseq X Ten platform according to the manufacturer's instructions.

### Phased variants calling and HLA typing with 10x Genomics linked-read data

10x Genomics linked-read data were first processed with proc10xG (process_10xReads.py v0.0.2, regen_10xReads.py v0.0.1, filter_10x-Reads.py v0.0.1) (https://github.com/CeciliaDeng/proc10xG) to extract gem barcodes, and then linked reads without barcodes were aligned to the Lambda sequence using Bwa mem (v0.7.15-r1140) with the default options. After alignment, unmapped reads were extracted by Seqkit (v.0.10.0) and further filtered so that: (1) the number of reads per barcode is >2 and; (2) the N base is not present in the barcode sequence. The remaining unmapped reads with high barcode quality were converted back to the 10x Genomics linked-read format using proc10xG, and aligned to the hg38 reference using Long Ranger (v.2.2.2) with gatk3.8 to call phased variants. To evaluate the performance of phased variants calling within the targeted MHC region, we calculated switching error rates and Hamming error rates by comparing our result to two golden standards (the GIAB (v.4.2.1)[21] and the Illumina Platinum Genomes[22]) for GM12878 using WhatsHap compare[51].

For HLA-typing, we split the 10x Genomics linked-read data into two haplotype-partitioned read sets based on their phasing information generated from Long Ranger, and HLA types for each haplotype were predicted separately using the HLA-VBseq (v2)[23] with the IMGT/HLA database (v3.44).

### Targeted pseudo-haplotype assembly of the MHC region with 10x Genomics linked-read data

We performed targeted pseudo-haplotype assembly of the MHC region with 10x Genomics linked-read data using the Supernova assembler (v2.1.1) as described previously[12] with modifications. Briefly, linked-read data were first aligned to the hg38 reference, and the barcodes with at least one read mapped within the targeted MHC region were kept. For each barcode, the number of on-target mapped reads within every 1-kb window was calculated, and the genomic position for each mapped read was recorded. In order to remove artifacts generated from barcode collisions, the barcode with a distance between adjacent mapped reads >40 kb were removed[52]. The barcodes with the number of mapped reads within a 160-kb window >10 were kept to facilitate the assembly of the targeted MHC region. The linked-read data within each 160-kb window were further assigned into 10-kb bins, i.e. 0–10, 10–20 kb, etc. For each bin, the barcodes with mapped read density above the $n$th percentile were removed, so that the final coverage in each bin is just below 70×. The final filtered barcodes within each 160-kb window were subsequently merged and used to extract linked-read data for constructing the targeted pseudo-haplotype assemblies using the Supernova assembler (v2.1.1) with the mkoutput option.

### Haplotype-resolved assembly of the targeted region with PacBio HiFi reads only

First, PacBio HiFi reads were filtered using Pbmarkdup (v1.0.0) (https://github.com/PacificBiosciences/pbmarkdup) to remove duplicated reads from amplification, and reads aligned to the targeted region of the hg38 were recruited using Minimap2 (v2.17). In order to remove the bias generated from amplification, down-sampling was performed as described below. The HiFi reads were assigned into 10-kb bins, and the reads in each bin with the shortest read length were removed to reach a final coverage of 60×. The primary assembly of the targeted region was generated from the remaining HiFi reads using Hifiasm (v0.11-r302)[5], and it was further polished using Racon (v1.4.20)[53] for two rounds to remove potential assembly errors. Then, HiFi reads aligned to the corrected primary assembly were recruited using Minimap2 and variants were called from the recruited HiFi reads using DeepVariants (v0.10.0)[54] and phased using WhatsHap (v0.17)[51]. Phased heterozygous variants, together with the corrected primary assembly, were used to separate the HiFi reads into two haplotype-partitioned read sets using WhatsHap. Finally, each haplotype-partitioned HiFi read set, together with untagged reads, was used to assemble two haplotypes of the targeted region using Hifiasm. For the targeted MHC region, we further aligned all HiFi reads to the two assembled haplotypes to identify supplementary HiFi reads (79 HiFi reads identified), which were missed from the initial alignment to the hg38 reference and these supplementary HiFi reads were combined with initial recruited HiFi reads to generate the final haplotype-resolved assemblies of the targeted MHC region.

### Haplotype-resolved assembly of the targeted region with both 10x Genomics linked-read data and PacBio HiFi reads

First, PacBio HiFi reads were filtered using Pbmarkdup (https://github.com/PacificBiosciences/pbmarkdup) to remove duplicated reads from amplification, and reads aligned to the targeted region of the hg38 were recruited using Minimap2. Then, variants were called from the recruited HiFi reads using DeepVariants[54], and further re-genotyped using WhatsHap[51]. A high-confidence set of heterozygous variants from HiFi reads were obtained by the intersection of heterozygous variants generated from DeepVariants and WhatsHap and then overlapped with phased variants within the targeted MHC region (chr6: 28903952-33268517) obtained from 10x Genomics linked-read data (Long Ranger) to generate a highly reliable set of heterozygous variants with phasing information. Finally, WhatsHap was used to separate the HiFi reads into two haplotype-partitioned read sets based on the collection of highly reliable phased heterozygous variants, together with the hg38 reference. In order to remove the bias generated from amplification, down-sampling was performed for each haplotype-partitioned read set as described below. The HiFi reads were assigned into 10-kb bins, and the reads in each bin with the shortest read length were removed to reach a final coverage of 30×. Each haplotype-partitioned HiFi read set, together with untagged reads, was used to assemble two haplotypes of the targeted region with Hifiasm[5]. Since an individual's MHC sequence can be quite different from the hg38 reference, HiFi reads from highly polymorphic regions may be missed from the initial step of read recruitment, resulting in gaps in the assembly result. To address this issue, we aligned all HiFi reads to the two assembled haplotypes to identify supplementary HiFi reads (68 HiFi reads identified), which were missed from the initial alignment to the hg38. These supplementary HiFi reads were combined with initially recruited HiFi reads to generate the final haplotype-resolved assembly of the targeted MHC region.

### Evaluation of haplotype-resolved assembly of the targeted region

To evaluate the completeness and accuracy of the haplotype-resolved assembly, two assembled haplotypes were aligned to the targeted region of the hg38, or the targeted region of genome assemblies for GM12878 cells reported recently[8] using Minimap2 (v2.17 and v2.24)[55].

The completeness of two assembled haplotypes was evaluated based on the alignment coverage and consistency, respectively. To evaluate the accuracy of our assembly, phased variants were called by aligning the phased contigs against the hg38 reference using Dipcall-1[56] with the following parameters: (1) xasm5 option removed to include regions with high divergence; (2) -z 200000,1000 to improve the contiguity of the alignment and; (3) -L 10000 to set the minimal length of the region to be 10 kb. Switching error rates and Hamming error rates were calculated by comparing our phased variants to the two-phased variant collections for GM12878 downloaded from the GIAB (v.4.2.1)[21] and the Illumina Platinum Genomes[22] using WhatsHap compare[51]. Hap.py (v0.3.15) (https://github.com/Illumina/hap.py) was used to evaluate variants with bed files from high-confidence regions[57]. For HLA typing, HLA alleles with 8-digit resolution were predicted by the direct alignment of two assembled haplotypes separately to the IMGT/HLA database (v3.44) (https://www.ebi.ac.uk/ipd/imgt/hla/) using Minimap2.

For Sanger sequencing of the *HLA-C* alleles, we first amplified the two targeted regions containing nucleotides different from the available *HLA-C* alleles in the IMGT/HLA database using nested PCR with primer sequences containing EcoRI or BamHI restriction enzyme digestion sites (Supplementary Data 7). The amplified DNA fragments were cloned into the pUC19 vector, and extracted plasmids from individually picked colonies were sequenced using Sanger sequencing.

### Collapsed analyses
Collapsed sequences were calculated as described previously[58] with modifications. Briefly, we first generated personal genome references, in which two targeted assembled personal MHC haplotypes were combined separately with the non-MHC region of the hg38. Then, we aligned downloaded PacBio HiFi reads of GM12878 cells from the GIAB (15; 20 kb) to each reference independently using Minimap2. The sequencing coverage was calculated using samtools (v1.9), and we defined collapsed bases as regions with at least 10 kb in length in the reference with the averaged coverage greater than expected coverage (mean + at least three standard deviations) of chromosome 6 excluding the targeted MHC region.

### Coordinates mapping between the targeted assembled personal haplotypes and the hg38 reference
To facilitate the direct comparison of two targeted assembled personal MHC haplotypes to the MHC region of the hg38 for genomics analyses, we generated genomic positions for both haplotypes based on their corresponding coordinates in the hg38 reference. Briefly, both haplotypes were aligned separately to the MHC region of the hg38 using MUMmer (v4.0.0beta2)[59]. The results were further processed using the delta-filter command with the option of "−1" to keep one-to-one alignment intervals, and sequences with the alignment format were generated using the show-aligns command. The final mapping coordinates were obtained by converting the alignment results to genomic coordinates using a customized perl script. The mapping coordinates between the two haplotypes were also defined similarly.

### Number of genetic variants and CpG variants within the targeted MHC region
For each haplotype, we counted the total number of genetic variants relative to the hg38 reference, which includes both SNPs and InDels, in each 10 kb window, and then plotted these counts sliding along the targeted MHC region. Similarly, we also counted the sum of CpG gains and losses relative to the hg38 in each 10 kb window and then drew the plot accordingly.

### Bisulfite sequencing data alignment and evidence extraction
The raw bisulfite sequencing reads were quality-controlled and then aligned separately to bisulfite-converted three references: 1. the non-MHC region of the hg38 combined with haplotype 1 of targeted assembled personal MHC haplotypes; 2. the non-MHC region of the hg38 with haplotype 2 of targeted assembled personal MHC haplotypes; and 3. the lambda phage genome sequence), using bowtie2 (v2.2.3)[60] with options -score-min L,0,0 for perfect alignment, while reads were mapped to the bisulfite-convert hg38 reference using bowtie2 with default options. After alignment, bismark (v0.22.1)[61] was used for evidence extraction to get methylated and unmethylated cytosines at each CpG site.

### Effect of CpG variants on DNA methylation analysis
Average DNA methylation levels for haplotype-specific or shared CpGs are estimated by aligning the sequencing reads to each personal genome reference of the corresponding targeted assembled personal MHC haplotype.

### Allele-specific methylation (ASM) analysis
Allele-specific methylation analysis was performed to identify differentially methylated regions between two haplotypes of the targeted MHC region of GM12878 cells. For ASM, we conducted two distinct analyses: one with haplotype-specific CpGs included and the other with haplotype-specific CpGs removed. For the analysis without haplotype-specific CpGs, only CpGs appearing at least twice per allele group were included in the analysis (this filtering step was not done for the analysis with haplotype-specific CpGs included, as the haplotype-specific CpGs would be excluded inevitably). After handling this haplotype-specific CpGs, the downstream analyses were the same, described as follows. We smoothed the evidence data using bsseq (v1.24.4)[62] with options: ns = 50, $h$ = 800, maxGap = $10^8$. Following smoothing, $t$-statistic was computed and DMRs were identified using a smooth window containing either 50 CpGs or a width of 800 bp, whichever was larger. Next, regions satisfying the following criteria were deemed putative DMRs: (1) a $t$-statistic with an absolute value >3.6; (2) must contain at least 3 CpG sites; and (3) must have a methylation difference of at least 10%.

To validate allele-specific methylation on the promoter region of *HLA-DPA1* gene, we performed bisulfite Sanger sequencing. Briefly, 500 ng of genomic DNA from GM12878 cells was converted with sodium bisulfite using the EZ DNA Methylation-Gold Kit (Zymo Research, Irvine, CA, USA), and samples were eluted in 20 µl of elution buffer according to the manufacturer's instructions. The targeted region containing three CpG sites within the DMR-DPA1 was amplified with nested PCR primers (Supplementary Data 7) with EcoRI or BamHI restriction enzyme digestion sites and subsequently cloned into the pUC19 vector. Plasmids were extracted separately from picked colonies and evaluated using Sanger sequencing.

### Luciferase reporter assay
The luciferase reporter assay was performed as described previously[63] with modifications. The CpG-free firefly luciferase reporter vector, pCpG-free basic vector (Invivogen), was used for detecting the promoter's activity. DNA containing eight repeats of a fragment in the DMR-DPA1 (TTACGTTA) were synthesized and then cloned into the promoter region of the pCpG-free basic vector upstream of the firefly luciferase gene. Methylation of the luciferase reporter was carried out using M.SssI CpG methyltransferase (NEB). Methylated or unmethylated firefly luciferase reporter was co-transfected with the control pRL-TK vector constitutively expressing Renilla luciferase into HEK293T cells using PEI. Cells were harvested 48 h post-transfection for luciferase activity using the dual-luciferase reporter assay system (Promega). Promoter activity was determined by calculating Firefly normalized against Renilla. All assays were performed in triplicates.

## Illumina Infinium EPIC methylation array data analysis

The raw EPIC idat files were preprocessed and normalized using minfi package v1.28.0[64] using default options. The methylation β value for each CpG site on the probe was calculated on the 0–1 scale $(M/(M + U + 100)$, where M and U represent the methylated and unmethylated signal intensities respectively). For the CpG sites on the array, we converted their genomic coordinates from the hg19 to the hg38 using the liftover (https://genome.ucsc.edu/cgi-bin/hgLiftOver) and overlapped their positions with those from bisulfite sequencing data to identify shared CpGs between the two platforms. Two groups of shared CpGs were determined by whether genetic variants are present within 5 bp of the interrogated CpG site on the probe of EPIC array, or absent from the probe. The methylation correlations between EPIC and bisulfite sequencing data were plotted for these two groups of shared CpGs accordingly, which were calculated by the coefficient of determination $R^2$ calculated from the linear regression of bisulfite sequencing data by EPIC array, and were equivalent to the square of Pearson's $R$.

## Analysis and validation of allele-specific expression

RNA-Seq data was generated from three replicates of GM12878 cells. Sequencing reads were quality-controlled using fastp (v0.21.0)[65], and filtered to remove adapter sequences, low-quality bases, as well as reads with read-length <50 bp using Trimmomatic (v0.39)[66]. The filtered reads were then aligned to the two personal genome references separately (the non-MHC region of the hg38 combined with haplotype 1 of targeted assembled personal MHC haplotypes; or the non-MHC region of the hg38 with haplotype 2 of targeted assembled personal MHC haplotypes) using HISAT2 (v2.1.0)[67] with default parameters. The rmdup_pe.py command of the WASP package (https://github.com/bmvdgeijn/WASP)[68] was used to remove PCR duplicated reads. For each alignment, only uniquely aligned reads were kept for quantification. By comparing the gene annotation file download from NCBI (ftp://ftp.ncbi.nlm.nih.gov/genomes/all/GCA/000/001/405/GCA_000001405.28_GRCh38.p13/GCA_000001405.28_GRCh38.p13_genomic.gff.gz) and mapping coordinates generated for two haplotypes by comparing to the hg38, gene annotation information was obtained for two assembled haplotypes.

The expression level of each gene was calculated separately for each allele using the bedtools multicov command. The reads of each gene were normalized using the estimateSizeFactors command in the DESeq2 package (v1.38.0)[69]. A two-sided Wald test $P$ value was calculated using DESeq2 for both alleles of each gene. Allele-specific expressed genes were identified with a statistical significance of adjusted $P$-value < 0.05.

Allele-specific expression of *HLA-DPA1* in GM12878 cells was validated by pyrosequencing, allele-specific qRT-PCR, and Sanger sequencing of amplified PCR clones. For pyrosequencing, the exon 3 region of the *DPA1* gene was PCR amplified from either genomic DNA or cDNA of GM12878 cells, and pyrosequencings were performed to calculate the ratio of targeted SNPs for both alleles from the amplification products using sequencing primers (Supplementary Data 7). For allele-specific qRT-PCR, primers (Supplementary Data 7) specific for each haplotype of exon 4 regions of *DPA1* were used to amplify the cDNA of GM12878 cells. The relative expression of each haplotype of the *DPA1* gene was calculated by normalizing its level to that of *GAPDH*. For Sanger sequencing of amplified PCR clones, the coding region of the *DPA1* gene containing several SNPs between two different haplotypes was amplified from the cDNA of GM12878 cells with PCR primers (Supplementary Data 7) with EcoRI or BamHI restriction enzyme digestion sites and subsequently cloned into the pUC19 vector. Plasmids were extracted separately from picked colonies and evaluated using Sanger sequencing.

## Reporting summary

Further information on research design is available in the Nature Portfolio Reporting Summary linked to this article.

## Data availability

All relevant data were generated from GM12878 cells. All relevant data supporting the key findings of this study are available within the article and its Supplementary Information file. The targeted assembled haplotypes and called variants can be downloaded from https://liulab.fudan.edu.cn/targeted_assemblies.html. DNA sequencing data with the Illumina short-read platform (accession code no. SRR17250933 for whole-genome sequencing; accession code no. SRR17250934 for the targeted enriched MHC region; accession code no. SRR21079439 for the targeted enriched *RHCE* and *CR1* regions), 10x Genomics linked-read (accession code no. SRR17250932 for the targeted enriched MHC region; accession code no. SRR21079437 for the targeted enriched *RHCE* region; accession code no. SRR21079438 for the targeted enriched *CR1* region), the PacBio HiFi long-read (accession code no. SRR17250931 for the targeted enriched MHC region; accession code no. SRR21079435 for the targeted enriched *RHCE* region; accession code no. SRR21079436 for the targeted enriched *CR1* region), and RNA-Seq data (accession code no. SRR17250928–SRR17250930, [https://www.ncbi.nlm.nih.gov/sra/?term=SRR17250929]) were deposited with Sequence Read Archive (SRA). Bisulfite sequencing data from the targeted enriched MHC region and the Illumina methylation EPIC beadchip data from genomic DNA of GM12878 cells are available in the Gene Expression Omnibus (GEO) database under accession numbers GSE192499 and GSE192501, respectively.

Assemblies and assembly-based phased variant calls from Garg et al. were acquired from [ftp://ftp.dfci.harvard.edu/pub/hli/whdenovo/]. PacBio HiFi reads used in collapsed analyses were acquired from the GIAB Consortium: [ftp://ftp-trace.ncbi.nlm.nih.gov/ReferenceSamples/giab/data/NA12878/HudsonAlpha_PacBio_CCS/PBmixSequel851_2_B01_PCCL_30hours_15kbV2PD_70pM_HumanHG001_CCS/m64109_200815_033514.fastq.gz], [ftp://ftp-trace.ncbi.nlm.nih.gov/ReferenceSamples/giab/data/NA12878/HudsonAlpha_PacBio_CCS/PBmixSequel851_1_A01_PCCL_30hours_15kbV2PD_70pM_HumanHG001_CCS/m64109_200813_162416.fastq.gz], [ftp://ftp-trace.ncbi.nlm.nih.gov/ReferenceSamples/giab/data/NA12878/HudsonAlpha_PacBio_CCS/PBmixSequel846_3_C01_PCDB_30hours_15kbV2PD_70pM_HumanHG001_CCS/m64109_200808_191025.fastq.gz], [ftp://ftp-trace.ncbi.nlm.nih.gov/ReferenceSamples/giab/data/NA12878/HudsonAlpha_PacBio_CCS/PBmixSequel846_1_A01_PCCL_30hours_15kbV2PD_70pM_HumanHG001_CCS/m64109_200805_204709.fastq.gz]; [ftp://ftp-trace.ncbi.nlm.nih.gov/ReferenceSamples/giab/data/NA12878/HudsonAlpha_PacBio_CCS/PBmixSequel846_4_D01_PBXM_30hours_21kbV2PD_70pM_HumanHG001_CCS/m64109_200810_062248.fastq.gz] and [ftp://ftp-trace.ncbi.nlm.nih.gov/ReferenceSamples/giab/data/NA12878/HudsonAlpha_PacBio_CCS/PBmixSequel846_2_B01_PCCM_30hours_21kbV2PD_70pM_HumanHG001_CCS/m64109_200807_075817.fastq.gz]. IMGT/HLA database (v3.44) was acquired from [https://www.ebi.ac.uk/ipd/imgt/hla/]. The source data underlying Figure. 1b, 4e, 4g and Supplementary Figure.7c, 8c, 866 8d are provided as a Source Data file. Source data are provided with this paper.

## Code availability

All the codes used in this work is available at: https://liulab.fudan.edu.cn/targeted_assemblies_codes.html

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

## Acknowledgements

This work was supported by grants from the National Natural Science Foundation of China (No. 82171837 to Yun Liu), the National Key R&D Program of China (No. 2021YFC2701001 to Yun Liu), Program of Shanghai Academic/Technology Research Leader Grant (No. 20XD1420400 to Yun Liu), Shanghai Municipal Science and Technology Major Project (Nos. 2017SHZDZX01 and 2018SHZDZX01 to Yun Liu) and ZJLab, and the Shanghai Science and Technology Foundation (20ZR1404700 to Yun Liu). We thank Qianjie Wang and Wilson Cheng from PacBio APAC for useful discussion, troubleshooting efforts, and analysis suggestions.

## Author contributions

Yun Liu conceived the idea. T.L. performed the experiments with assistance from J.M., W.M., Z.J., H.Y., J.W., S.D., S.S., W.Y., and B.L. D.D., D.Z., and Yicheng Lin performed the analyses with assistance from M.Z. and Z.C. H.L., Z.Z., Y.J., and Z.X. suggested experimental design. Yun Liu and W.Q. supervised the work. T.L., W.Q. and Yun Liu wrote the manuscript with assistance from all authors.

## Competing interests

A patent application has been filed based on the findings in this paper (Yun Liu et al. CN201911027231.0).
