## [Peer Review File · Nature Communications]

CRISPR-based targeted haplotype-resolved assembly of a megabase regionREVIEWER COMMENTS

Reviewer #1 (Remarks to the Author):

This paper proposes a method for the targeted haplotype-resolved genome assembly using the CRISPR-based enrichment, and gives an example for the major histocompatibility complex (MHC) region. The authors thoroughly evaluate the impact of the targeted haplotype-resolved assembly for the downstream analysis. Although the authors present a good method for the community, I have major and fundamental concerns that should be addressed before this manuscript is suitable for publication.

1. To evaluate the quality of the targeted haplotype-resolved genome assembly, the authors took the dipasm assembly of HG002 as the ground truth. However, there is a significantly better benchmark for the MHC region of HG002 (see Chin, C.S. et al. "A diploid assembly-based benchmark for variants in the major histocompatibility complex". *Nat Commun* 11, 4794 (2020)). The author should use this benchmark to evaluate their results.

2. The authors said, "...targeted approach achieved comparable completeness and accuracy..." For repeat-rich regions like MHC, a very important metric is how many repeats are collapsed in assemblies. For example, the authors could adopt the solution in Extended Data Fig. 9 of "Automated assembly of high-quality diploid human reference genomes", which is the bake-off paper of the Human Pangenome Reference Consortium.

3. With the Pacbio HiFi reads and the new assembly algorithms like HiCanu and hifiasm, it is not hard to assemble MHC. For example, the Human Pangenome Reference Consortium (HPRC) has recently released the haplotype-resolved assemblies for >40 human samples, and most of these assemblies entirely resolve the whole MHC or just have one breakpoint. I would recommend the authors to demonstrate the advantages of their method in some even harder regions. The GIAB Consortium has recently released a curated variation benchmark for challenging medically relevant autosomal genes (see Wagner, Justin, et al. "Curated variation benchmarks for challenging medically relevant autosomal genes." *Nature Biotechnology* (2022)). It is helpful if the authors could apply their method on at least one resolved gene and one unresolved gene by GIAB.

4. The authors claim their method is de novo, but it is not since it still requires the reference genome to call heterozygous variants. To make it de novo, the authors should at least follow the dipasm solution, which calls variants based on the primary assembly using the HiFi reads from the same sample, instead of the Grch38 (see Garg, Shilpa, et al. "Chromosome-scale, haplotype-resolved assembly of human genomes." *Nature biotechnology* 39.3 (2021)). This solution is still not ideal as it is more likely to

collapse repeats. It would be good to utilize graph-based algorithms like hifiasm, to directly assemble two haplotypes from the assembly graph. Hifiasm has `-3/-4` option to support it.

5. The authors said, "...targeted approach achieved comparable completeness and accuracy with greatly reduced computing complexity..." However, there is no result showing the running time/memory requirement of their method. It is hard to say if their computing complexity is really reduced as their computational pipeline includes multiple heavy steps.

Reviewer #2 (Remarks to the Author):

The manuscript describes an approach to enrich long DNA from the MHC region and do linked read and long read sequencing to get haplotype-resolved assembly. This is a useful proof of principle for targeting important, challenging regions with long reads, but I have a few suggestions for improving the description of the methods and benchmarking.

1. For benchmarking variant calls, the authors compare to GIABv3.3.2, which excludes the challenging regions of the MHC. The newer v4.2.1 benchmark includes variants from an assembly-based benchmark, which should better highlight the strengths of their approach:

<https://doi.org/10.1101/2020.07.24.212712>

2. Since the segmental duplication including important genes C4A/B, TNXA/B, and CYP21A2 can be challenging to assemble, it would be good to check variant accuracy particularly in these genes

3. Mapping reads from RNAseq and bisulfite seq to the diploid assembly is challenging to do in a robust way. It would be useful to explain the rationale behind their methods more clearly and include this in the main text. Particularly, it would be good to explain why the differential expression and methylation are not related to mapping artifacts to a diploid assembly

4. The authors conclude, "It can be safely assumed that other functional genomics analyses of any targeted genomic regions would benefit from our approach of using haplotype-resolved assembly of targeted regions as the personal reference." I don't think this is generally true of most of the genome. Rather, it would be better to say challenging regions like the MHC could benefit from this approach.

5. Based on the high false positive rate, Table 1 doesn't appear to use benchmark bed files when doing the comparison, as is strongly recommended by both GIAB and PG. Please do the comparison only inside the benchmark bed files, using a tool like hap.py (<https://github.com/Illumina/hap.py>), as recommended in the GA4GH best practices - <https://rdcu.be/bqpDT>

6. This method would be much more valuable if it did not require both linked reads and long reads. Could the authors try a hifiasm assembly without using 10x for phasing?

7. The targeting method appears to result in uneven coverage, which would be good to state in the main text clearly.

8. What causes breaks in the assemblies? How many are related to the targeting probes, or regions that weren't phased, or repetitive sequences, or something else?

9. Are the assemblies and corresponding variant calls available from the analyses? I could not easily find these

Reviewer #3 (Remarks to the Author):

Li et al describe a method for targeted sequencing and assembly of individual haplotypes from a diploid cell line. As a proof-of-principle, the method uses CRISPR sgRNAs to isolate high molecular weight DNA of the ~4Mb major histocompatibility (MHC) locus in GM12878 cells, followed by 10X Genomics linked-read and PacBio HiFi sequencing. The sequencing data is combined to assemble both haplotypes. By comparing their assemblies to an existing haplotype-resolved assembly of the GM12878 cell line and the hg38 reference genome the authors showed their haplotype assemblies were accurate. Finally, they detect allele-specific methylation and expression patterns of their assemblies of the MHC compared to hg38.

A method capable of generating accurate and targeted sequencing of individual haplotypes without requiring trio data is a useful tool. An additional strength of this targeted approach is one potential application of this method would be to combine sets of sgRNAs for multiplexed sequencing of multiple loci of interest to simultaneously resolve structural variation at many loci. However, the technical challenges and labor involved in the technique make it unlikely to be widely adopted. To increase the utility of the technique the authors might describe how others may adapt and apply it to other complex regions. Garg et al were able to assemble accurate haplotype assemblies of entire chromosomes, including the MHC haplotypes from the same cell line, so the targeted nature of the authors approach is the main strength of this manuscript and more focus should be directed to the advantages of their method as opposed to whole genome sequencing. The below concerns should be addressed to help strengthen the manuscript and the utility of their method.

Major concerns:

1. The authors argue that its significance is in the reduced cost and the potential scaling of the method to sequence many genomes to understand structural variation of the same locus across a population. While accurate haplotype assemblies are crucial for understanding variable regions like the MHC, if reduced cost truly is a benefit of this approach there is not a quantification of this claim (a comparison to Garg et al may be most appropriate) and accurate quantification of labor and time involved is

difficult. Please include calculations justifying the reduced cost of their strategy as compared to whole genome PacBio HiFi sequencing employed by Garg et al.

2. The authors argue that this method could be applied to population studies, however, at a population level, variation of the MHC would require the sequencing of many genomes from many derived cell lines. Due to the labor of massive amounts of cell culture and technical challenge required to isolate high molecular weight DNA from agarose reliably, the resources and labor required of this technically demanding preparation prior to sequencing makes this method unlikely to be scaled for use in population studies. The authors should revise the text to indicate this limitation.

3. Can the authors generate a single, haplotype-resolved, contiguous sequence of the entire MHC region from the PacBio HiFi sequencing alone? It is unclear from the text whether this is achievable as the authors combine the 10X and PacBio data to generate the two haplotype assemblies of MHC. If the authors can generate a single contig for each haplotype, this would make this method more widely used.

4. The authors should explicitly state that their strategy is dependent on having a reference sequence and some understanding of the intervening duplicated sequences. They used sgRNAs within the MHC region to half the 4.7Mb region into two ~2Mb segments. If the MHC region is duplicated, inverted, or deleted, then this would change the CRISPR-excised segment sizes and would have to be detectable on a PFG, which can be challenging. Defining these limitations in the text will be helpful for others who wish to pursue such a strategy.

5. Figure 4c shows the comparison of each haplotype to hg38, both its identity shared and the identity not shared (which is redundant). Notably, there is no comparison of methylation of haplotype 1 versus haplotype 2, which would be of greater biological interest.

Minor concerns:

1. Please explain why haplotype 2 seems to have more bisulfite sequencing data than haplotype 1 in Figure 4f.

2. Since this is largely a methods paper it would be important to show all of the details. For example, the pulse field gel image of the CRISPR excised band would be important. It would also be good to have the range of DNA yields they observed as limiting DNA will be an important consideration in downstream analyses.

3. Figure 3 takes up essentially an entire page, only to depict several small indels in a small portion of the MHC locus, which does not warrant an entire figure. I would recommend removing or moving to the supplement.

4. For Figure 4g it would be helpful to see the differential methylation levels of the DMR-DPA1 region specifically in GM12878 since that was the cell line the sequencing was generated from, and not HEK293T cells, which likely have a different methylation pattern.

5. In the Figure 4b legend more annotation would be nice, I assume that DP,DQ,DR are genes, but this should be more explicit.

6. Line 675, define "edit distance"?

Point-by-point response to reviewers' comments (NCOMMS-22-03567)

Reviewer #1

This paper proposes a method for the targeted haplotype-resolved genome assembly using the CRISPR-based enrichment, and gives an example for the major histocompatibility complex (MHC) region. The authors thoroughly evaluate the impact of the targeted haplotype-resolved assembly for the downstream analysis. Although the authors present a good method for the community, I have major and fundamental concerns that should be addressed before this manuscript is suitable for publication.

Response: We thank the reviewer for the recognition of the importance of our work.

Comment #1: To evaluate the quality of the targeted haplotype-resolved genome assembly, the authors took the dipasm assembly of HG002 as the ground truth. However, there is a significantly better benchmark for the MHC region of HG002 (see Chin, C.S. et al. "A diploid assembly-based benchmark for variants in the major histocompatibility complex". Nat Commun 11, 4794 (2020)). The author should use this benchmark to evaluate their results.

Response: We thank the reviewer for the suggestion to evaluate our results with a better benchmark. Chin, C.S. *et al.* used the 10x Genomics Linked-reads and long reads (Oxford Nanopore and PacBio HiFi) to generate two MHC haplotypes of HG002 and establish an assembly-based variant benchmark. HG002 is a sample from the GIAB Consortium, also named as NA24385. However, the sample we used in this work is NA12878 (also known as HG001 in the GIAB), which makes it unsuitable for direct comparison.

Recently, the GIAB Consortium released a new benchmark for NA12878 (Wagner *et al.*, Cell Genomics, 2022), which used accurate linked and long reads to expand benchmark variants including challenging genomic regions such as the MHC. In light of this, we now revised the manuscript by evaluating our targeted haplotype-resolved MHC assembly to this new better benchmark, GIAB v.4.2.1. As shown in the revised Table 1, we identified 30100 genetic variants within the targeted MHC region with false negative rates (FNR) lower than 2.8%. All variants were phased, with both switch error rates and Hamming error rates lower than 0.01% (revised Supplementary Table 1). Importantly, with the new benchmark, the false discovery rates (FDR) were reduced from ~61.9% (compared to the GIAB v3.3.2) to ~19.3% (compared to the GIAB v4.2.1), indicating an expanded benchmark variants in the MHC region. We also compared the whole-genome haplotype-resolved assemblies of NA12878 from Garg *et al.* to the new benchmark, and observed comparable results (Table 1 and Supplementary Table 1). We have now revised the manuscript accordingly.

Comment #2: The authors said, " ... targeted approach achieved comparable completeness and accuracy..." For repeat-rich regions like MHC, a very important metric is how many repeats are collapsed in assemblies. For example, the authors could

adopt the solution in Extended Data Fig. 9 of “Automated assembly of high-quality diploid human reference genomes”, which is the bake-off paper of the Human Pangenome Reference Consortium.

Response: We thank the reviewer for the valuable comment. As suggested by the reviewer, we downloaded the HiFi reads of GM12878 from the GIAB and estimated collapsed sequences by analyzing HiFi read coverage across the whole targeted MHC region (Supplementary Fig. 4c), similar to what was done in “Automated assembly of high-quality diploid human reference genomes” (bioRxiv: 2022.03.06.483034; doi: <https://doi.org/10.1101/2022.03.06.483034>). We identified 0 kb and 20 kb collapsed sequences for each of our two haplotypes. In contrast, the whole-genome assembly result from Garg *et al.* has 30 kb and 20 kb collapsed sequences for each haplotype (Supplementary Table 8).

In addition, we also looked closely to the region where collapsed sequences are mainly located at from both assemblies. This region contains important C4A/B, CYP21A2 and TNXA/B genes known with segmental duplications. 20 kb of collapsed repeats were observed in this region of haplotype 1 from Garg *et al.*, while there is no evidence of collapsed repeats from our targeted approach (Fig. 3b, Supplementary Table 8). However, 20 kb of collapsed repeats were observed for the haplotype 2 of both assemblies. Consistently with these, when comparing the assembly results to the bed files of this region from both the GIAB and the Illumina Platinum Genomes benchmarks, F1 scores were higher than 0.83 for our targeted result, while they were lower than 0.6 in Garg *et al.* (Supplementary Table 9). All these results highlight that our targeted approach can achieve high level of accuracy, comparable to the whole-genome assembly result. We have now included this in the revised manuscript, as follows:

Results (page 11):

“In order to evaluate the assembly accuracy of complex regions with repetitive content which have traditionally been collapsed or excluded, we estimated number of base pairs that are collapsed in the assembly results by analyzing raw HiFi read coverage of the whole targeted MHC region. Compared to the whole-genome assembly result, our targeted approach yields similar or less collapsed repeats for both assembled haplotypes (Supplementary Fig. 4c, Supplementary Table 8). Further evaluation of segmental duplication region with important genes C4A/B, TNXA/B and CYP21A2 showed that 20 kb of collapsed repeats were observed in this region of haplotype 1 from Garg *et al.*, while there is no evidence of collapsed repeats from our targeted approach (Fig. 3b, Supplementary Table 8). Consistently, when comparing the assembly results to the bed files of this region from both benchmarks, F1 scores were higher than 0.83 for our targeted result, while they were lower than 0.6 in Garg *et al.* (Supplementary Table 9). All these results highlight that our targeted approach can achieve high level of accuracy, comparable to the whole-genome assembly result.”

Comment #3: With the Pacbio HiFi reads and the new assembly algorithms like HiCanu and hifiasm, it is not hard to assemble MHC. For example, the Human Pangenome Reference Consortium (HPRC) has recently released the haplotype-

resolved assemblies for >40 human samples, and most of these assemblies entirely resolve the whole MHC or just have one breakpoint. I would recommend the authors to demonstrate the advantages of their method in some even harder regions. The GIAB Consortium has recently released a curated variation benchmark for challenging medically relevant autosomal genes (see Wagner, Justin, et al. “Curated variation benchmarks for challenging medically relevant autosomal genes.” Nature Biotechnology (2022)). It is helpful if the authors could apply their method on at least one resolved gene and one unresolved gene by GIAB.

Response: We appreciate the reviewer’s recommendation to demonstrate the advantages of our method in some even harder regions. We have now applied our method for one GIAB-resolved gene (RHCE) and one GIAB-unresolved gene (CR1), reported by Justin, et al., Nature Biotechnology, 2022. For both regions, we performed CRISPR-based targeted enrichment (Supplementary Fig. 8) followed by targeted haplotype-resolved assembly using either the PacBio HiFi reads only or combined it with the 10x Genomics linked-reads. Other than the HiFi only assembly of CR1 gene, both assembly strategies generated haplotype-resolved assemblies of the two targeted genes with only 1 contig for each haplotype (Supplementary Table 12). We observed high level of accuracy for our targeted assembly results by comparing called variants to the two benchmarks (the GIAB v.4.2.1 and the Illumina PG) (Supplementary Table 13 and Supplementary Table 14), as well as observing the presence of supporting PacBio HiFi reads across the targeted region (Supplementary Fig. 9). All the results demonstrated that, similar to haplotype-resolved whole-genome assemblies, our targeted approach can be used to resolve and characterize genetic variants in other genomic complex regions. We have now included this in the result section of the revised manuscript (page 17-18).

Comment #4: The authors claim their method is de novo, but it is not since it still requires the reference genome to call heterozygous variants. To make it de novo, the authors should at least follow the dipasm solution, which calls variants based on the primary assembly using the HiFi reads from the same sample, instead of the Grch38 (see Garg, Shilpa, et al. “Chromosome-scale, haplotype-resolved assembly of human genomes.” Nature biotechnology 39.3 (2021)). This solution is still not ideal as it is more likely to collapse repeats. It would be good to utilize graph-based algorithms like hifiasm, to directly assemble two haplotypes from the assembly graph. Hifiasm has `3/-4` option to support it.

Response: We thank the reviewer for the suggestion. We agreed with the reviewer that our method is not de novo since the hg38 reference is used. In fact, the reference sequence was used in two steps of our targeted assembly approach: 1. It was used to recruit HiFi reads aligned to the targeted region of the hg38; and 2. It was used to call phased heterozygous variants. For the second step, we were able to remove the use of the hg38 reference now by using the solution suggested by the reviewer. Briefly, we utilized hifiasm to construct the primary assembly from the HiFi reads aligned to the targeted region of the hg38. Variants were called from HiFi reads aligned to the primary

assembly using Minimap2 and DeepVariants, and were further phased using WhatsHap. Then, WhatsHap was used to separate the HiFi reads into two haplotype-partitioned read sets based on phased heterozygous variants. Eventually, each haplotype-partitioned HiFi read set, together with untagged reads, was used to assemble two haplotypes of the targeted region using hifiasm. In this strategy, we not only removed the use of the hg38 reference in the variant calling step, but also removed the reliance on the 10x Genomics linked-read data (Supplementary Fig. 3c).

With this PacBio HiFi only strategy, our targeted MHC assembly result contains 6 contigs for haplotype 1 and 4 contigs for haplotype 2, and covers more than 96% of the targeted MHC region for both haplotypes (Supplementary Table 4). To evaluate the accuracy of the assembly result, called variants were compared to the two benchmarks (the GIAB v4.2.1 and the Illumina PG). Even though most of genetic variants were identified with FNRs lower than 3% (Table 1) and switch error rates were lower than 0.5% (Supplementary Table 1), suggesting a relatively high level of accuracy of the assembly result, we did observe a considerable number of heterozygous variants were called with inaccurate phasing information (Table 1) with Hamming error rates higher than 45% (Supplementary Table 1). This is, in fact, the result of a single switch error in the middle of the targeted assembly result (Supplementary Figure 3d).

We also applied this PacBio HiFi only strategy to two challenging medically relevant genes, RHCE and CR1. For the GIAB-resolved RHCE genes, both the PacBio HiFi only strategy and the strategy with both HiFi data and 10x Genomic linked-read generated highly consistent assembly results with high accuracy (Supplementary Table 13 and Supplementary Table 14). However, for the GIAB-unresolved CR1 gene, the assembly result from the PacBio HiFi only strategy contains one breakpoint for one of the haplotypes with Hamming errors as high as 40%. In contrast, the assembly result from the strategy with both PacBio HiFi data and 10x Genomic linked-read data contains only 1 contig for each haplotype, with high level of accuracy of both switch errors and Hamming errors lower than 2%. This result is similar to what we have been observed for the targeted assembly of the MHC region, and suggest that even though we are able to generate a haplotype-resolved assembly of the targeted region with relatively good quality by using the PacBio HiFi data only, the completeness and phasing accuracy of the targeted assembly can still be improved by combining with another strand-specific sequencing technology for some complex regions, such as the MHC and CR1.

Even though the newly-added PacBio HiFi only strategy did not use the reference sequence for the variant calling step, we still used the hg38 reference initially in order to recruit HiFi reads aligned to the targeted region of the reference. Thus, our approach is still not *de novo*. Therefore, we removed the description of “*de novo*” in the revised manuscript and acknowledged the reliance on the reference sequence for our targeted approach in the discussion section of the revised manuscript, as follows:

Discussion (page 20):

“Moreover, even though generating targeted haplotype-resolved assembly from the PacBio HiFi reads with Hifiasm is independent from the reference genome, the

reference sequence is still needed in order to filter the HiFi reads aligned to the targeted region.”

Comment #5: The authors said, “...targeted approach achieved comparable completeness and accuracy with greatly reduced computing complexity...” However, there is no result showing the running time/memory requirement of their method. It is hard to say if their computing complexity is really reduced as their computational pipeline includes multiple heavy steps.

Response: We apologize for this omission. We have now compared the running time/peak memory usage of our method to Garg et al, as well as two additional haplotype-resolved assembly methods (Koren, S. et al., Nature Biotechnology, 2018 and Chin, C.S. et al., Nat Communication, 2020). Koren, S. et al. used the trio-binning approach to *de novo* assemble the haplotype-resolved genomes of HG001, and Chin, C.S. et al. used long reads (PacBio HiFi and ONT) and linked reads to assemble the MHC region of HG002 from whole-genome sequencing (WGS) data. By using the same machine with 30 CPU threads, our targeted approach run for 25-26 wall-clock hours with the peak memory around 30 Gb. Chin, C.S. et al. was about 3 times slower than ours, with the peak memory around 103 Gb. Garg et al. was about 5 times slower than ours, with the peak memory around 91 Gb. Koren S et al. was the slowest with the highest peak memory. All these have been summarized in Supplementary Table 16, and been added into the discussion section of the revised manuscript (page 19).

Reviewer #2

The manuscript describes an approach to enrich long DNA from the MHC region and do linked read and long read sequencing to get haplotype-resolved assembly. This is a useful proof of principle for targeting important, challenging regions with long reads, but I have a few suggestions for improving the description of the methods and benchmarking.

Response: We thank the reviewer for a positive and thorough review of our manuscript.

Comment #1: For benchmarking variant calls, the authors compare to GIABv3.3.2, which excludes the challenging regions of the MHC. The newer v4.2.1 benchmark includes variants from an assembly-based benchmark, which should better highlight the strengths of their approach:<https://doi.org/10.1101/2020.07.24.212712>

Response: We thank the reviewer for the suggestion to evaluate our results with the GIAB v4.2.1 benchmark. We have now replaced the evaluation results from using the GIAB v3.3.2 to using the GIAB v4.2.1 and found that the false positive rates (FPR) were reduced from ~61.9% (v3.3.2) to ~19.3% (v4.2.1). We have now revised manuscript accordingly.

Comment #2: Since the segmental duplication including important genes C4A/B, TNXA/B, and CYP21A2 can be challenging to assemble, it would be good to check variant accuracy particularly in these genes.

Response: We thank the reviewer for this valuable suggestion. We have now evaluated the variant accuracy of this region by comparing called variants to the bed files from two benchmarks (the GIAB v.4.2.1 and the Illumina PG). As illustrated in the Supplementary Table 9, all the SNPs in this region curated in the GIAB were identified in our targeted assembly result and phased correctly. The comparison to the Illumina PG also showed high level of accuracy (F1 score: 0.948) with a relatively low switch error (3.51%) and hamming error rates (1.72%). We also evaluated whole-genome assembly result from Garg *et al.* and found that F1 scores were lower than ours (less than 0.6) with higher switch error and hamming error rates (Supplementary Table 9). We have now included this in the revised manuscript, as follows:

Results (page 11):

“Further evaluation of segmental duplication region with important genes C4A/B, TNXA/B and CYP21A2 showed that 20 kb of collapsed repeats were observed in this region of haplotype 1 from Garg *et al.*, while there is no evidence of collapsed repeats from our targeted approach (Fig. 3b, Supplementary Table 8). Consistently, when comparing the assembly results to the bed files of this region from both benchmarks, F1 scores were higher than 0.83 for our targeted result, while they were lower than 0.6 in Garg *et al.* (Supplementary Table 9).”

In addition, in addressing the **Comment #2** from **the Reviewer #1**, we also evaluated collapsed sequences of the whole targeted MHC region, as well as the region containing important C4A/B, TNXA/B, and CYP21A2 genes. As mentioned above, the results showed that there are less collapsed sequences in our targeted assembly when compared to the whole-genome assembly of Garg *et al.*, further highlighting the high level of accuracy in our targeted approach.

Comment #3: Mapping reads from RNAseq and bisulfite seq to the diploid assembly is challenging to do in a robust way. It would be useful to explain the rationale behind their methods more clearly and include this in the main text. Particularly, it would be good to explain why the differential expression and methylation are not related to mapping artifacts to a diploid assembly.

Response: We thank the reviewer for highlighting this important issue. We have now added a paragraph of clarification to the manuscript in the result section (page 13), together with a figure illustration (Supplementary Fig. 5b), as follows: “It has been proposed that the use of personal genome as the reference is a solution to deal with alignment-related artifacts for both gene expression^{30, 31} and DNA methylation analyses³². In light of this, we replaced the hg38 reference with personal genome references, in which two targeted assembled personal MHC haplotypes was combined separately with the non-MHC region of the hg38. Short-read sequencing data is then aligned to each personal genome reference individually to limit the effects of genetic variation on sequence alignment (Supplementary Fig. 5b).”

Even though our approach of replacing a single reference genome with personal genome can improve the accuracy of alignment quantifications, as demonstrated similarly by other works, we agreed with the reviewer that mapping reads to the diploid assembly is quite challenging to do and our approach still did not address other issues,

such as aligning sequencing reads to multiple genome locations or to both personal genome references, which may lead to mapping artifacts. Thus, we added a paragraph of discussing (page 21), as follows: “With that being said, identification of ASE and ASM events accurately is quite challenging to do with short-read sequencing data. Even though our approach can limit the effects of alignment bias caused by genetic variations, other challenges, such as sequencing reads aligned to multiple genome locations or to both personal genome references, remain to be resolved. Nevertheless, it can be safely assumed that other functional genomics analyses of other challenging regions like the MHC would benefit from our approach of using haplotype-resolved assembly of targeted regions as the personal reference.”

Comment #4: The authors conclude, “It can be safely assumed that other functional genomics analyses of any targeted genomic regions would benefit from our approach of using haplotype-resolved assembly of targeted regions as the personal reference.” I don’ t think this is generally true of most of the genome. Rather, it would be better to say challenging regions like the MHC could benefit from this approach.

Response: We agreed with the reviewer and has revised the manuscript as suggested.

Comment #5: Based on the high false positive rate, Table 1 doesn’t appear to use benchmark bed files when doing the comparison, as is strongly recommended by both GIAB and PG. Please do the comparison only inside the benchmark bed files, using a tool like hap.py (<https://github.com/Illumina/hap.py>), as recommended in the GA4GH best practices - <https://rdcu.be/bqpDT>.

Response: We appreciate the reviewer’s valuable comment to use benchmark bed files when doing the comparison. As suggested, we reassessed the accuracy of called variants from our targeted assembly and Garg *et al.* by doing the comparison using the bed files from two benchmarks (the GIAB v.4.2.1 and the Illumina PG) with hap.py. As demonstrated on Supplementary Table 7, when compared to the Illumina PG, our targeted assembly outperformed Garg *et al.* on all the recall rates and precision rates for both SNPs and Indels, with the F1 scores over 0.975 (SNPs) and 0.891 (Indels). When compared to the GIAB, even though our recall rate for Indels is a little bit lower than Garg *et al.* (0.958 vs. 0.961), our targeted assembly showed better results for all the other recall rates and precision rates than those from Garg *et al.*, with the F1 scores over 0.996 (SNPs) and 0.944 (Indels). These results demonstrate that our targeted assembly has high accuracy on variants calling of the high-confidence regions. We have now included this in the revised manuscript, as follows:

Results (page 9):

“Evaluation of variants using bed files from high-confidence regions showed that F1 scores (representing a harmonic mean of precision and recall) were higher than 0.89 for both benchmarks (Supplementary Table 7).”

Comment #6: This method would be much more valuable if it did not require both linked reads and long reads. Could the authors try a hifiasm assembly without using

10x for phasing?

Response: We truly appreciated the reviewer's recommendation and agreed that the method would be much more valuable if it can work without 10x Genomics linked-reads. In addressing **Comment #4** from **the Reviewer #1**, we showed that we can now use PacBio HiFi reads only to achieve haplotype-resolved assembly of the targeted MHC region. The new assembly result showed high coverage of the targeted MHC region (more than 96% for both haplotypes), and can be used to identify most of genetic variants (FNRs lower than 3%). While switch error rates were lower than 0.5% (Supplementary Table 1), a considerable number of heterozygous variants were called with inaccurate phasing information (Table 1) with Hamming error rates higher than 45% (Supplementary Table 1). A direct comparison to Garg et al. showed that this is the result of a single switch error in the middle of the targeted MHC region (Supplementary Figure 3d). Similar result was observed for the GIAB-unresolved challenging medically relevant gene, CR1. These results suggests that, even though we were able to generate a haplotype-resolved assembly of the targeted genomic complex region with relatively good quality by using the PacBio HiFi data only, the completeness and phasing accuracy of the targeted assembly can still be improved by combining with another strand-specific sequencing technology, such as the 10x Genomics linked-reads. We have now included these in the revised manuscript.

Comment #7: The targeting method appears to result in uneven coverage, which would be good to state in the main text clearly.

Response: We thank the reviewer for highlighting the lack of this information. As suggested, we have now included the following sentence to the manuscript in the result section (page 6), as follows: "Notably, the coverage within the entire 2.3 Mb DNA fragment is relatively the same but quite different between two fragments (Fig. 1c), suggesting that the enrichment of the targeted region is dependent on the cleavage efficiency of corresponding sgRNAs."

Comment #8: What causes breaks in the assemblies? How many are related to the targeting probes, or regions that weren't phased, or repetitive sequences, or something else?

Response: We checked all the breakpoints in the assemblies, and, as demonstrated in Supplementary Table 6, we observed zero or almost zero PacBio HiFi reads aligned to the breakpoints. Since before the construction of sequencing library, enriched high molecular weight DNA was amplified using the Ultra-Low DNA Input Workflow (PacBio), we believe these breakpoints may be the result of the absence of HiFi reads from biased amplification. We have now included these in the revised manuscript, as follows:

Results (page 9):

"We did observe several breakpoints in our assembly result (Supplementary Table 6), and they seem to be resulting from the absence of HiFi reads (Supplementary Fig. 4a, Supplementary Table 6) from biased amplification."

Comment #9: Are the assemblies and corresponding variant calls available from the analyses? I could not easily find these.

Response: We apologize for the omission and have now included the assemblies and corresponding variant calls at: https://liulab.fudan.edu.cn/targeted_assemblies.html. We have revised the “Data Availability” section in the revised manuscript (page 37).

Reviewer #3

Li et al describe a method for targeted sequencing and assembly of individual haplotypes from a diploid cell line. As a proof-of-principle, the method uses CRISPR sgRNAs to isolate high molecular weight DNA of the ~4Mb major histocompatibility (MHC) locus in GM12878 cells, followed by 10X Genomics linked-read and PacBio HiFi sequencing. The sequencing data is combined to assemble both haplotypes. By comparing their assemblies to an existing haplotype-resolved assembly of the GM12878 cell line and the hg38 reference genome the authors showed their haplotype assemblies were accurate. Finally, they detect allele-specific methylation and expression patterns of their assemblies of the MHC compared to hg38. A method capable of generating accurate and targeted sequencing of individual haplotypes without requiring trio data is a useful tool.

Response: We thank the reviewer for a positive and thorough review of our manuscript.

An additional strength of this targeted approach is one potential application of this method would be to combine sets of sgRNAs for multiplexed sequencing of multiple loci of interest to simultaneously resolve structural variation at many loci. However, the technical challenges and labor involved in the technique make it unlikely to be widely adopted. To increase the utility of the technique the authors might describe how others may adapt and apply it to other complex regions. Garg et al were able to assemble accurate haplotype assemblies of entire chromosomes, including the MHC haplotypes from the same cell line, so the targeted nature of the authors approach is the main strength of this manuscript and more focus should be directed to the advantages of their method as opposed to whole genome sequencing.

Response: We thank the reviewer for pointing out an additional strength of our targeted approach. As suggested by the reviewer and in addressing **Comment #3** from **the Reviewer #1**, to expand the application of our targeted approach, we have now applied our method to two challenging medically relevant autosomal genes (RHCE and CR1), which were recently described by the GIAB (Justin, et al., Nature Biotechnology, 2022). These two gene regions were isolated from GM12878 cells either alone or together by different combination of sgRNA sets (Supplementary Fig. 8a, Supplementary Fig. 8b). Both qPCR analyses (Supplementary Fig. 8c, 8d) and sequencing with the Illumina short-read platform (Supplementary Fig. 8e) showed that both targeted regions were successfully enriched. Next, we performed haplotype-resolved assembly of the two regions using either the PacBio HiFi reads only or combined it with the 10x Genomics linked-reads. Other than the HiFi only assembly of CR1 gene, both assembly strategies

generated haplotype-resolved assemblies of the two targeted genes with only 1 contig for each haplotype (Supplementary Table 12). Comparing called variants to the two benchmarks (the GIAB v4.2.1 and the Illumina PG) illustrates the high level of accuracy of the assemblies (Supplementary Table 13 and Supplementary Table 14). As suggested by the reviewer, our application on other complex regions further demonstrate the generality of our targeted approach. We have now included these in the result section of the revised manuscript (page 17-18).

The below concerns should be addressed to help strengthen the manuscript and the utility of their method.

Major concerns:

Comment #1: The authors argue that its significance is in the reduced cost and the potential scaling of the method to sequence many genomes to understand structural variation of the same locus across a population. While accurate haplotype assemblies are crucial for understanding variable regions like the MHC, if reduced cost truly is a benefit of this approach there is not a quantification of this claim (a comparison to Garg et al may be most appropriate) and accurate quantification of labor and time involved is difficult. Please include calculations justifying the reduced cost of their strategy as compared to whole genome PacBio HiFi sequencing employed by Garg et al.

Response: We apologize for this omission. We have now compared the cost of our method to Garg et al and two additional haplotype-resolved assembly methods (Koren, S. et al., Nature Biotechnology, 2018 and Chin, C.S. et al., Nat Communication, 2020). Koren, S. et al. used the trio-binning approach to *de novo* assemble the haplotype-resolved genomes of HG001, and Chin, C.S. et al. used long reads (PacBio HiFi and ONT) and linked reads to assemble the MHC region of HG002 from whole-genome sequencing (WGS) data. As now summarized in Supplementary Table 16, for each sample, the cost of our targeted method is ~ 2900 USD, roughly 22% of Garg et al (~ 13000 USD) and Chin, C.S. et al (~ 13000 USD). The trio-binning approach used by Koren, S. cost the most with ~ 28000 USD. One thing needs to be pointed out is that sequencing is the major source of the cost (~ 90%). For targeted regions shorter than the MHC (such as the two challenging medically relevant genes, RHCE and CR1), the cost for our targeted approach will fall down accordingly as fewer sequencing data is needed.

Comment #2: The authors argue that this method could be applied to population studies, however, at a population level, variation of the MHC would require the sequencing of many genomes from many derived cell lines. Due to the labor of massive amounts of cell culture and technical challenge required to isolate high molecular weight DNA from agarose reliably, the resources and labor required of this technically demanding preparation prior to sequencing makes this method unlikely to be scaled for use in population studies. The authors should revise the text to indicate this limitation.

Response: We thank the reviewer for this comment. Even though we only demonstrated our targeted approach using GM12878 cells, it has been showed that CRISPR-based

strategy for the enrichment of HMW DNA molecules can be similarly applied to primary tissues, including peripheral blood mononuclear cells (Gabrieli, T. et al., *Nucleic Acids Res*, 2018) and breast tumors (Gilpatrick, T. et al., *Nat Biotechnol*, 2020). We also tested this on several human PBMC samples and found high enrichment of the targeted MHC region. Thus, the application of our targeted approach to population studies will not require the generation of many derived cell lines, making it feasible to be scaled for use in population studies. We have now included this in the discussion section of the revised manuscript (page 19).

Comment #3: Can the authors generate a single, haplotype-resolved, contiguous sequence of the entire MHC region from the PacBio HiFi sequencing alone? It is unclear from the text whether this is achievable as the authors combine the 10X and PacBio data to generate the two haplotype assemblies of MHC. If the authors can generate a single contig for each haplotype, this would make this method more widely used.

Response: We truly appreciated the reviewer's recommendation. As we addressed in **Comment #6** from **the Reviewer #2**, we have now used PacBio HiFi reads only to achieve haplotype-resolved assembly of the targeted MHC region.

Comment #4: The authors should explicitly state that their strategy is dependent on having a reference sequence and some understanding of the intervening duplicated sequences. They used sgRNAs within the MHC region to half the 4.7Mb region into two ~2Mb segments. If the MHC region is duplicated, inverted, or deleted, then this would change the CRISPR-excised segment sizes and would have to be detectable on a PFG, which can be challenging. Defining these limitations in the text will be helpful for others who wish to pursue such a strategy.

Response: We agreed and thank the reviewer for pointing out this limitation, and have now added this in the discussion section of the revised manuscript, as follows:

Discussion (page 19-20):

“An issue that may complicate studies adopting our targeted approach is that the enrichment strategy of designing sgRNAs is dependent on the understanding of the targeted region, as well as the sequences flanking it. If the targeted region is duplicated, inverted, or deleted, this would change the molecular weight of enriched DNA molecules, making it difficult to be recovered from pulsed field gel.”

Comment #5: Figure 4c shows the comparison of each haplotype to hg38, both its identity shared and the identity not shared (which is redundant). Notably, there is no comparison of methylation of haplotype 1 versus haplotype 2, which would be of greater biological interest.

Response: We agreed with the reviewer and have now compared the methylation of haplotype 1 versus haplotype 2. Similar to what we observed previously, when the different haplotype was used as the reference, the methylation level of haplotype-specific CpGs was extremely different (Fig. 4c).

Minor concerns:

Comment #1: Please explain why haplotype 2 seems to have more bisulfite sequencing data than haplotype 1 in Figure 4f.

Response: Figure 4f shows the bisulfite Sanger sequencing result of picked colony clones generated from amplified PCR products of the DMR-DPA1 region. We randomly picked a total of 39 colonies, and the sequencing data revealed that 15 of them came from haplotype 1, and 24 from haplotype 2. Even though the number of identified colonies coming from haplotype 2 is more than that from haplotype 1, this difference is not statistically significant (p value = 0.3675, Fisher's exact test), indicating it is potentially caused by chance.

Comment #2: Since this is largely a methods paper it would be important to show all of the details. For example, the pulse field gel image of the CRISPR excised band would be important. It would also be good to have the range of DNA yields they observed as limiting DNA will be an important consideration in downstream analyses.

Response: We apologize for this omission. We have now added the pulse field gel images of the CRISPR excised bands of all targeted regions (Supplementary Fig. 2a, Supplementary Fig. 8a, 8b), as well as the range of DNA yields (Supplementary Table 15), in the revised manuscript.

Comment #3: Figure 3 takes up essentially an entire page, only to depict several small indels in a small portion of the MHC locus, which does not warrant an entire figure. I would recommend removing or moving to the supplement.

Response: We thank the reviewer for this suggestion. We have now moved the original Fig. 3c to Supplementary Fig. 4d., and added a new figure illustrating the analysis result of collapsed repeats comparing our targeted assemblies to the whole-genome assemblies from Garg *et al.* (the new Fig. 3b).

Comment #4: For Figure 4g it would be helpful to see the differential methylation levels of the DMR-DPA1 region specifically in GM12878 since that was the cell line the sequencing was generated from, and not HEK293T cells, which likely have a different methylation pattern.

Response: We thank the reviewer for this comment. The differential methylation level of the DMR-DPA1 region in GM12878 was validated using bisulfite Sanger sequencing in Figure 4f. For Figure 4g, we wanted to examine whether the differential methylation of the DMR-DPA1 region has any regulatory effects on gene expression *in vitro*. Thus, the DNA sequences of the DMR-DPA1 region was synthesized and cloned into the promoter region of CpG-free firefly luciferase reporter vector. The methylation level of this plasmid was manipulated *in vitro* with the treatment of DNA methyltransferase, M.SssI. Then, methylated, or unmethylated plasmids were transfected into HEK293T cells and the effects on gene expression were evaluated by the luciferase assay. The

reason that HEK293T cells were used here is that it has much better transient transfection efficiency comparing to the GM12878 cells, and the endogenous methylation level of the DMR-DPA1 region in HEK293T cells has no effect on the luciferase assay.

Comment #5: In the Figure 4b legend more annotation would be nice, I assume that DP,DQ,DR are genes, but this should be more explicit.

Response: We have modified the Figure 4b legend with more detailed gene annotations.

Comment #6: Line 675, define “edit distance”?

Response: “Edit distance” has now been defined in the Figure 2b legend.

REVIEWER COMMENTS

Reviewer #1 (Remarks to the Author):

The authors have addressed all of my concerns. The revised manuscript is ready for publication.

Reviewer #2 (Remarks to the Author):

The authors have done a good job of responding to the reviews, but I have 2 additional items that would be good to address:

1. The authors state: "While the current deployment of the PacBio HiFi platform for genome assemblies requires large amounts of starting materials (e.g., more than 25 µg of genomic DNA from diploid human cells, in order to generate a minimum of 80 Gb data required for whole-genome assemblies), we were able to generate a 12 kb HiFi library with at least 60× coverage of the targeted MHC region (Supplementary Fig. 3b) from 20 ng of enriched HMW DNA" To make this a better comparison, it is important to know the amount of DNA used prior to the enrichment as well.
2. I think the additional analysis of challenging medically relevant genes is potentially really useful. However, it is not clear to me what the benchmark is for RHCE and CR1 when the authors calculate FNRs, since the difficult regions of these genes were not well-resolved in GIABv4.2.1 or PG. Supp Fig 9 shows that there are breaks in the alignments of the assemblies to the interesting challenging parts of both genes, and it is not clear whether these are due to breaks in the assemblies or breaks in the alignments. The GIAB CMRG benchmark didn't include CR1 because there was a break in the alignment of the assembly for the older minimap2 version used in that work. In our experience, minimap2 v2.24+ will now usually align across the entire genes. It doesn't appear the authors specified the version used, but it would be good to try the latest version for this figure at least.

Reviewer #3 (Remarks to the Author):

Overall, the authors addressed our major concerns with the manuscript. The addition of two non-MHC regions from which the authors were able to construct single-contig assemblies was a valuable addition to the manuscript. The authors also addressed the concern of population-level scaling of this method by testing PBMCs, although the data for this claim is not in the revised manuscript.

However, several minor concerns still remain:

1. The added RHCE and CR1 regions bolster the manuscript's rigor. However, the qPCR analyses which confirm specificity of the CRISPR-excised region used genes far from the enriched regions of interest. Is there any reason the genes immediately flanking these regions were not analyzed? It seems it would be preferable to ensure accurate cutting by using neighboring genes to the enriched region.
2. The luciferase assay does validate the differential methylation of each haplotype. This assay reveals that, when demethylated, this region is associated with increased luciferase activity, indicating an increase in expression. However, this assay was performed in HEK293T cells. While the authors addressed this concern by mentioning the higher transfection efficiency making HEK293T cells preferable to GM12878 cells for such an assay, it should be noted in the text that follow-up experiments are needed confirm this pattern is biological and holds in other cell types as well. The authors should explain a reason why they did not perform allele-specific qRT-PCR in the GM12878 cells for a more direct comparison of the methylation pattern's relation to expression in the actual cell type which was sequenced?
3. The legend of Figure 1a is very long and the text and be condensed. The last sentence would be more appropriate in the discussion section.
4. In lines 387-391 of the manuscript, the authors have addressed the concern of being unable to scale this technique to the population level due to the involved cell culture work by mentioning the technique's applicability to PBMCs. However, breast tumor-derived cells would NOT address this concern.
5. Since this manuscript is in part development of a new method it is good to show all steps. Thus Ideally a gel image, pre-CRISPR excision, would be good to show together with the post-excision gel image.

Point-by-point response to reviewers' comments (NCOMMS-22-03567)

Reviewer #1

The authors have addressed all of my concerns. The revised manuscript is ready for publication.

Response: We were grateful to the reviewer for the constructive suggestions and were delighted to hear that we addressed all of his/her concerns and the manuscript is ready for publication.

Reviewer #2

The authors have done a good job of responding to the reviews, but I have 2 additional items that would be good to address:

Response: We thank the reviewer for the recognition of the improvement of the manuscript.

Comment #1: The authors state: "While the current deployment of the PacBio HiFi platform for genome assemblies requires large amounts of starting materials (e.g., more than 25 µg of genomic DNA from diploid human cells, in order to generate a minimum of 80 Gb data required for whole-genome assemblies), we were able to generate a 12 kb HiFi library with at least 60× coverage of the targeted MHC region (Supplementary Fig. 3b) from 20 ng of enriched HMW DNA" To make this a better comparison, it is important to know the amount of DNA used prior to the enrichment as well.

Response: We thank the reviewer for this valuable suggestion. We have now added this information in the result section (page 6) and the corresponding method section (page 24), as follows: "Four agarose plugs equal to a total of 4×10^6 cells were used for the isolation of the MHC region.", and "Four agarose plugs were used for the MHC isolation (~20 µg of genomic DNA), and two plugs for the medically relevant genes (RHCE and CR1) (~10 µg of genomic DNA)."

Comment #2: I think the additional analysis of challenging medically relevant genes is potentially really useful. However, it is not clear to me what the benchmark is for RHCE and CR1 when the authors calculate FNRs, since the difficult regions of these genes were not well-resolved in GIABv4.2.1 or PG.

Response: We thank the reviewer for highlighting the lack of clarity on this. For both RHCE and CR1, we called phased variants from the assemblies of the entire targeted regions and then compared them to the GIABv4.2.1 and the Illumina PG benchmarks to calculate FNRs. We have clarified this in the revised result section (page 18). We also agreed with the reviewer that the difficult regions of both RHCE and CR1 genes were not well-resolved in the GIAB or PG, thus they are not ideal benchmarks. In recognizing of this, we added the following sentence in the revised manuscript: "Since challenging medically relevant regions of RHCE and CR1 genes were not well-resolved

in both benchmarks, we identified many new variants, resulting a high level of FDRs (more than 40%) (Supplementary Table 13)." (page 18)

Supp Fig 9 shows that there are breaks in the alignments of the assemblies to the interesting challenging parts of both genes, and it is not clear whether these are due to breaks in the assemblies or breaks in the alignments. The GIAB CMRG benchmark didn't include CR1 because there was a break in the alignment of the assembly for the older minimap2 version used in that work. In our experience, minimap2 v2.24+ will now usually align across the entire genes. It doesn't appear the authors specified the version used, but it would be good to try the latest version for this figure at least.

Response: We thank the reviewer for this valuable comment. As suggested, we have now compared the alignment results with different versions of minimap2 (v2.17 vs. v2.24) (Supplementary Fig. 10). As correctly pointed out by the reviewer, the latest version of minimap2 v2.24 can align across the entire CR1 gene and two large deletions (17,095 bp and 18,555 bp) in the haplotype 1 of CR1 gene were identified (Supplementary Fig. 10). Thus, the break for the older version of minimap2 (v2.17) in the haplotype 1 of CR1 gene is due to the alignment. We have revised the result section, as follows:

Results (page 18-19):

"with some breaks in the alignments of the assemblies to the hg38 reference (Supplementary Fig. 9). Recently, a new version of minimap2 (v2.24) was released, which improves alignment strategies around long poorly aligned regions. In order to distinguish whether these breaks are due to breaks in the assemblies or in the alignments, the alignment results were further compared with different versions of minimap2 (Supplementary Fig. 10). While no change was observed for RHCE gene, two large deletions (17,095 bp and 18,555 bp) were identified in the haplotype 1 of CR1 gene using the latest version of minimap2 (v2.24) (Supplementary Fig. 10)."

Additionally, we added the version information for all the software used in this work in the method section of the revised manuscript.

Reviewer #3

Overall, the authors addressed our major concerns with the manuscript. The addition of two non-MHC regions from which the authors were able to construct single-contig assemblies was a valuable addition to the manuscript. The authors also addressed the concern of population-level scaling of this method by testing PBMCs, although the data for this claim is not in the revised manuscript.

Response: We thank the reviewer for the recognition of our effort and a thorough review of our manuscript.

However, several minor concerns still remain:

Comment #1: The added RHCE and CR1 regions bolster the manuscript's rigor. However, the qPCR analyses which confirm specificity of the CRISPR-excised region used genes far from the enriched regions of interest. Is there any reason the genes

immediately flanking these regions were not analyzed? It seems it would be preferable to ensure accurate cutting by using neighboring genes to the enriched region.

Response: There is no specified reason for the selection of flanking regions for the qPCR analyses, and we agreed with the reviewer that the neighboring genes would be a better choice for these analyses. We have now performed the qPCR analyses on the immediately flanking genes and updated the results accordingly (Supplementary Fig. 8c, 8d).

Comment #2: The luciferase assay does validate the differential methylation of each haplotype. This assay reveals that, when demethylated, this region is associated with increased luciferase activity, indicating an increase in expression. However, this assay was performed in HEK293T cells. While the authors addressed this concern by mentioning the higher transfection efficiency making HEK293T cells preferable to GM12878 cells for such an assay, it should be noted in the text that follow-up experiments are needed confirm this pattern is biological and holds in other cell types as well.

Response: We agreed with the reviewer and has added this information in the revised discussion section, as follows:

Discussion (page 21):

"but also revealed allele-specific transcriptional regulation of the *HLA-DPA1* gene through the luciferase assay. As the luciferase assay was performed in the HEK293T cells, it should be noted that further experiments are needed to confirm this pattern is biological and holds in other cell types as well."

The authors should explain a reason why they did not perform allele-specific qRT-PCR in the GM12878 cells for a more direct comparison of the methylation pattern's relation to expression in the actual cell type which was sequenced?

Response: We thank the reviewer for this suggestion. As suggested, we performed allele-specific qRT-PCR in the GM12878 cells (Supplementary Fig. 7b). In addition, we also performed Sanger sequencing of amplified PCR clones of the *HLA-DPA1* gene in the GM12878 cells (Supplementary Fig. 7c). Both results showed that the haplotype 2 of *DPA1* gene is significantly overexpressed compared to the haplotype 1 in the GM12878 cells. We have added this in the revised result section, as follows:

Results (page 16):

"This ASE was validated using pyrosequencing (Fig. 4e), allele-specific qRT-PCR (P-value = 0.0074, paired student's t-test, two-sided) (Supplementary Fig. 7b) and Sanger sequencing of amplified PCR clones (P-value = 0.0068, Fisher's exact test) (Supplementary Fig. 7c), and all of them showed that the haplotype 2 of *DPA1* gene is significantly overexpressed compared to the haplotype 1 in GM12878 cells."

Comment #3: The legend of Figure 1a is very long and the text and be condensed. The last sentence would be more appropriate in the discussion section.

Response: We agreed with the reviewer and has condensed the legend of Figure 1a as suggested.

Comment #4: In lines 387-391 of the manuscript, the authors have addressed the concern of being unable to scale this technique to the population level due to the involved cell culture work by mentioning the technique's applicability to PBMCs. However, breast tumor-derived cells would NOT address this concern.

Response: We agreed with the reviewer that using tumor-derived cells will involve a lot of work. However, in the referred work (ref. 13) by Gilpatrick, T. et al. (Nat Biotechnol, 2020), they showed in Fig.1b and Fig.2c that, in addition to tumor-derived cells, they can isolate HMW DNA molecules with CRISPR-based enrichment strategy on fresh breast tumor tissues. We hope this could address your concern.

Comment #5: Since this manuscript is in part development of a new method it is good to show all steps. Thus Ideally a gel image, pre-CRISPR excision, would be good to show together with the post-excision gel image.

Response: We agreed with the reviewer and the gel images pre-CRISPR excision were added (Supplementary Fig. 2a, Supplementary Fig. 8a), together with the post-excision gel images (Supplementary Fig. 2b, Supplementary Fig. 8b).